# Dome patterns in pelagic size spectra reveal strong trophic cascades

Axel G. Rossberg [1,2,3]*, Ursula Gaedke[4] & Pavel Kratina [1]*

In ecological communities, especially the pelagic zones of aquatic ecosystems, certain body-size ranges are often over-represented compared to others. Community size spectra, the distributions of community biomass over the logarithmic body-mass axis, tend to exhibit regularly spaced local maxima, called "domes", separated by steep troughs. Contrasting established theory, we explain these dome patterns as manifestations of top-down trophic cascades along aquatic food chains. Compiling high quality size-spectrum data and comparing these with a size-spectrum model introduced in this study, we test this theory and develop a detailed picture of the mechanisms by which bottom-up and top-down effects interact to generate dome patterns. Results imply that strong top-down trophic cascades are common in freshwater communities, much more than hitherto demonstrated, and may arise in nutrient rich marine systems as well. Transferring insights from the general theory of non-linear pattern formation to domes patterns, we provide new interpretations of past lake-manipulation experiments.

[1] School of Biological and Chemical Sciences, Queen Mary University of London, Mile End Rd, London E1 4NS, UK. [2] Centre for Environment, Fisheries and Aquaculture Science (Cefas), Pakefield Rd, Lowestoft NR33 0HT, UK. [3] International Initiative for Theoretical Ecology, Unit 10, 317 Essex Road, London N1 2EE, UK. [4] Department of Ecology and Ecosystem Modeling, Institute for Biochemistry and Biology, University of Potsdam, Am Neuen Palais 10, 14469 Potsdam, Germany. *email: axel@rossberg.net; p.kratina@qmul.ac.uk

Size spectra and food chains belong to the many concepts ecologists have invoked in order to understand structure and dynamics of complex ecological communities. The common conceptualisation of communities as simple food chains[1–4] formed by groups of organisms assigned to discrete trophic levels predicts existence of top-down trophic cascades[2–7], where pressures on top predators propagate to lower trophic levels, leading to alternating increases and decreases of biomass along the food chain. Such cascades are well documented[8]. Their strengths, however, vary considerably among study systems[6]. Differences in nutrient supply might contribute to this variation, in particular in pelagic ecosystems, where nutrients are known to play a crucial ecological role. This idea, however, has remained controversial. A review from 2010 presented evidence in support of three competing hypotheses[9]: that pelagic cascades are strongest at the highest nutrient concentrations, that they are strongest at intermediate nutrient concentrations, and that cascade strength is largely independent of nutrient availability and primary production. More recent reviews[10,11] do not see nutrient supply amongst the factors affecting cascade strengths. This is surprising, since increasingly sophisticated food-chain models have predicted stronger cascades for more productive systems[4,12,13]. Does this discrepancy indicate inherent limitations of food-chain theory?

An alternative way of looking at the structure of an ecological community is its size spectrum, the distribution of community biomass over the logarithmic body size axis[14–18], which can span a factor $>10^{14}$ in body mass (Fig. 1a). Size-spectrum theory underlies the use of size-based indicators in status assessments of aquatic ecosystems[19]. In pelagic size spectra one often observes distinct body-size ranges of high biomass density[20], known as domes[21], and depleted troughs between domes (Fig. 1a, b). The question what causes these domes remains subject of ongoing speculation[22–26], but the most common explanation invoked in recent empirical literature[27–29] and reviews[15,17] implies that the domes represent subsequent members of the aquatic food chain[21]. The dome structure arises through a bottom-up cascade, where the position and height of each dome is controlled by that to its left on the size axis[30,31]. For the left-most dome, another explanation would be required.

Here, we provide evidence for an alternative interpretation of domes, which, by combining ideas from food-chain and size-spectrum theory, resolves open questions pertinent to both. We demonstrate that domes are manifestations of top-down trophic cascades, enhanced by eutrophication (Fig. 1c). From the size-spectrum perspective, the result is a pronounced periodic modulation of the density of community biomass (large domes, deep troughs) along the logarithmic body size axis that becomes weaker with lower nutrient availability[32] (smaller domes and shallower troughs) and, in the oligotrophic open ocean, might disappear entirely[14,33] (Fig. 1a, b). At intermediate to high nutrient concentrations, however, non-linear effects can lead to deviations from classical food-chain theory in how size-spectra respond to pressures.

Self-organised periodic (i.e., regularly spaced) patterns, such as stripe patterns on animal skins, are a common natural phenomenon. It has long been speculated that such structures can arise not only in physical space but also in abstract spaces, for example, when competition generates patterns in ecological trait spaces[34–41]. However, any attempt to explain periodic patterns in nature must be mindful that general mathematical principles alone already predict periodic patterns as a common phenomenon[42]. Indeed, size-spectrum models can generate modulations through a variety of mechanisms[22–25,43]. Echoing a similar conundrum surrounding regularities in planetary orbits[44], determination of how domes arise in nature therefore requires more detailed agreement between data and model than the mere fact

that modulations are found. Insufficient demonstration of such agreement for existing process-based models is the reason why the dome pattern remains enigmatic. The need for detailed comparisons also limits what can be learned from minimal food-chain[4,13] or size-spectrum[22,45] models, despite their important role in illuminating basic mechanisms. Our comparison of model and data therefore considers specific patterns in the responses of size spectra and domes to nutrient enrichment.

Our working model, the non-linear Species Size Spectrum Model (SSSM), is designed to incorporate crucial elements of ecological realism while preserving mathematical simplicity and computational efficiency. This permits us to overcome limitations of earlier modelling approaches[16–18]. Using this model, we identify the mechanisms controlling the dependence of dome structure on nutrients, thus permitting us to delineate the conditions required for these patterns to arise.

## Results and Discussion

**The non-linear SSSM.** We found that a generalisation of the linear SSSM[25] to a full, non-linear model is capable of reproducing the observed phenomenology of domes (Fig. 1). The SSSM belongs to a family of similar models[24,46,47] going back to the Fish Community Size-Resolved Model[48–50]. Structure and motivation of the SSSM are best understood in the wider context of size-spectrum theory[16].

In the simplest dynamic size-spectrum models, individuals interacting in a community are distinguished only by their body masses[22,23,45]. New individuals are added at a constant rate or abundance at the lower end of the modelled size range. Individuals then grow by feeding on each other (or on small planktonic organisms that are not explicitly modelled) and converting the biomass of their prey into their own—at some given efficiency. These feeding interactions lead to corresponding predation mortality. The strength of feeding interactions is controlled by a predator–prey mass-ratio window function; it depends only on the body masses of predator and prey. Life-history parameters and physiological rates scale allometrically with body size. The resulting size-spectrum dynamics is described by a flux-divergence equation for the density of individuals along the size axis, known as the McKendrick—von Foerster Equation[22,51].

More advanced models[16,48] distinguish indiviuals not only by body mass but also by species identity, permitting these models to give a full account of the reproductive cycle. Each species has an associated maturation body mass, above which indiviuals invest a large proportion of food intake into reproduction rather than ontogenetic growth. The reproductive investment determines the rate at which individuals of each species are added at the lower end of the modelled size range of that species. In this model class, community dynamics are described by a system of coupled McKendrick—von Foerster Equations, one for each species. Any species' population biomass can grow or decline, and species experiencing low food availability and high predation mortality throughout their life cycles can go extinct.

Indeed, when modelling feeding interactions as depending only on the body masses of predators and prey, very few species coexist in these models[52]. A natural way to overcome this limitation is to acknowledge that, while body mass is an important trait affecting feeding interactions, the combined influence of all other traits can be even stronger[53,54]. In models this is implemented by either equipping species with abstract secondary traits[54,55] (usually assigned at random) that affect the strength of feeding interactions[56,57], or by directly multiplying the size-dependent interaction strength with a non-negative factor sampled at random for each species pair[48,50]. Both approaches

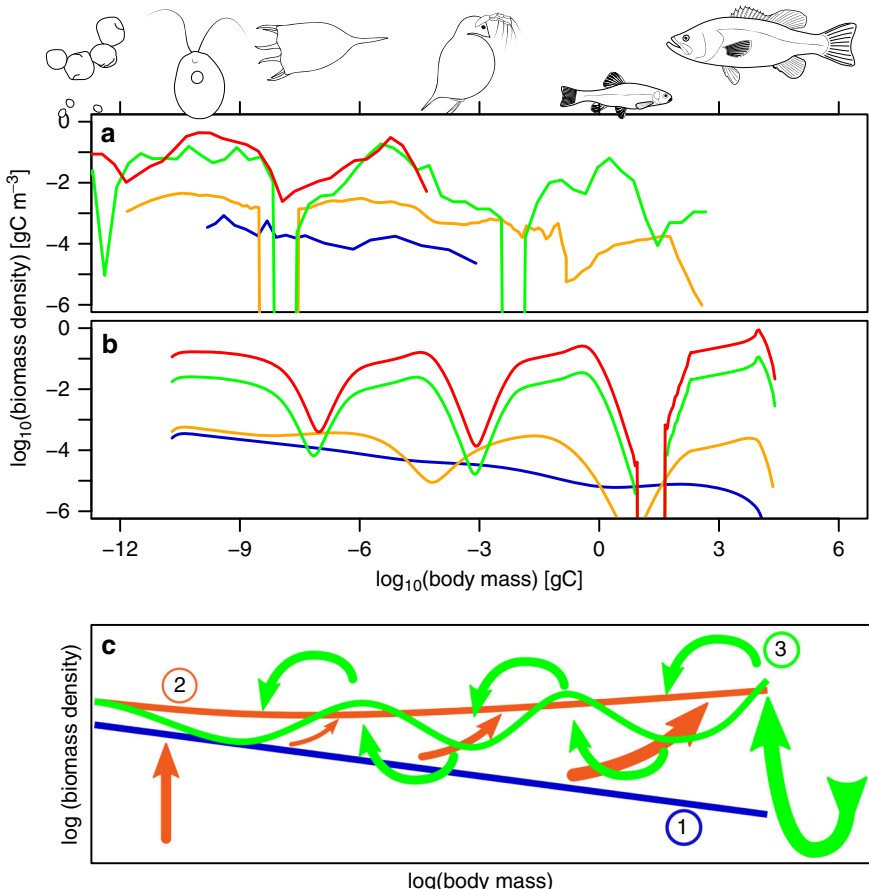

**Fig. 1** Illustration of dome formation in size spectra. Empirical size spectra in panel **a** are, in order of increasing nutrient enrichment (lowest to highest line), from the North Pacific[33] (blue), Lake Superior[27] (orange), Lake Ontario[86] (green) and Lake Müggelsee[83] (red). See Methods for the representation of size spectra used. Comparable simulation data in panel **b** are steady states of the Species Size Spectrum Model (SSSM) with varying eutrophication parameter $x$ (lowest to highest line; blue: $x = 1$, orange: $10^{0.2} = 1.58$, green: $10^{1.6} = 39.8$, red: $10^{2.4} = 251$). Panel **c** illustrates the causal chain generating domes in the SSSM. ① At low nutrient supply, the size-spectrum forms an approximate straight line on double-logarithmic axes (blue line). ② Increasing nutrient supply increases abundance of primary producers, and, through bottom-up trophic amplification, induces an upward-bending of the entire size-spectrum (orange line and arrows). ③ As a result, consumers become more satiated and more abundant relative to their resources, which both acts to amplify top-down cascades[4,13]. Low-predation mortality of the top predators induces such a cascade. This leads to formation of several domes along the size spectrum (green line and arrows)

lead to food-web models in which species of similar size can have very different prey and predators, implying reduced competition and a lower likelihood of competitive exclusion. While in these models many species can coexist, the original simplicity of the size-spectrum approach is lost. Species-rich models of this type become computationally expensive.

An important observation made studying such models is that the community size spectrum, obtained by adding the size spectra of all modelled species, is dominated for any given size class by species with maturation body sizes close to this size class[48]—consistent with analytic theory[25,51] and field data[58,59]. Structure and dynamics of community size spectra over large size ranges are therefore determined predominantly by variations in the population biomasses of species of different sizes, rather than by variations in intraspecific size structure[59,60].

The SSSM builds on this observation. Using an analytic technique called quasi-neutral approximation[61] (QNA), the species-level McKendrick—von Foerster Equations are replaced by a system of coupled ordinary differential equations for the dynamics of the species' population biomasses. The QNA is a powerful technique that permits us to retain implicitly descriptions of individual-level processes. In particular, the SSSM captures the Type II functional responses of individuals to food

availability from the underlying size-structured food-web model[48], and so the possibility of consumer satiation.

Because the SSSM aims to describe species-rich communities in a computationally efficient way, species are thought of as being grouped into narrow maturation body mass classes, revealing the distribution of community biomass over the logarithmic maturation body mass axis: the *species size spectrum*. The SSSM primarily models the dynamics of this species size spectrum. From this, the community size spectrum is reconstructed following the QNA methodology. However, as a results of lumping species into maturation body mass classes, their differentiation by secondary traits, found to be essential for food-web models of size-structured populations, gets lost. To counteract the resulting artificial competitive exclusion dynamics for species of similar size, the SSSM contains a phenomenological correction term[25], which accounts for differentiation of species by secondary traits. The structure of this term is similar to the intraspecific competition terms included in Lotka–Volterra models for niche differentiation along a single trait axis[37], but in the SSSM it is constructed such as to avoid unaccounted biomass losses.

Rather than explicitly distinguishing between primary producers and consumers in size spectra, the SSSM imposes a lower

boundary condition that fixes the abundances of the smallest species modelled. Nutrient enrichment is modelled by scaling these abundances by a eutrophication parameter $x$, assumed to be proportional to chlorophyll-$a$ concentration. Model equations are listed in Supplementary Note 1 together with a detailed discussion, including our correction to account for secondary traits. Details of simulation methods are described in Supplementary Note 3. The code we used to simulate the SSSM is provided as Supplementary Software 1.

**Basic model behaviour**. Model simulations with an ecologically plausible parameterisation (Supplementary Note 2) demonstrate that nutrient enrichment and the resulting increases in $x$ and overall community biomass do indeed result in strong dome formation (Fig. 1b).

As a simple demonstration that dome formation in this model is governed by a top-down cascading effect, we removed species from two domes at opposite ends of the size spectrum. First, we simulated harvesting of species from the dome with the largest body sizes, by increasing their mortality. This substantially reduced the size-spectrum modulation along the entire body size axis (Fig. 2a), as expected for a top-down cascade. If the domes would represent subsequent members of a food chain, the naive expectation would be that release from predation allows the intermediate dome to rise when the right-most dome is suppressed, but this is not what we found in simulations.

Second, we harvested species from the dome at the smallest body size. This removed that dome, but had little effect on the modulation of rest of the size spectrum (Fig. 2b, red dashes). The effect it had on larger-bodied individuals is explained by the overall removal of biomass from the system; it is identical to that resulting from a reduction of the eutrophication parameter from $x = 1.58$ to $1.38$ (Fig. 2b, blue dots).

**Comparison with data**. In addition to generating domes in process-based simulations, our model reproduces much of the rich phenomenology associated with nutrient enrichment in pelagic ecosystems, providing further strong support for the theory. To demonstrate this, we compiled 25 high-quality size spectrum data sets from the empirical literature. We only included data sets spanning at least six orders of magnitude in body mass that provided volumetric biomass density measures and where trophic status has been quantified as total phosphorus (TP) or chlorophyll-$a$ concentration (which we then expressed as TP using a published regression). To allow quantitative comparison of size spectra across studies, we expressed them in units of gram carbon[62] and as biomass densities along the ln(body mass)-axis[25,33]. Visual inspection of this data (Fig. 1a, Supplementary Note 4) confirms previous reports[62] that, on double-logarithmic axes, pelagic size spectra exhibit a linear relation that tends to be overlayed with a secondary structure of uniformly spaced domes. To quantify these features, we fitted both empirical and simulated size spectra to a combination of a linear relation and sinusoidal modulation of the form

$$
\begin{aligned}
\log_{10}(\text{biomass density}) = {} & S \log_{10}\left(\frac{\text{body mass}}{1\mu\text{gC}}\right) + B_0 \\
& + A \sin\left[\frac{2\pi \log_{10}(\text{body mass})}{D} - P\right].
\end{aligned}
\tag{1}
$$

The parameter $S$, representing the overall slope of the size spectrum, indicates to what extent community biomass is dominated by small organisms ($S < 0$) or large organisms ($S > 0$). Sheldon's hypothesis[14], that biomass is roughly equally distributed over all logarithmic body size classes, corresponds to $S = 0$. The parameter $B_0$ is the intercept at $1\,\mu\text{gC}$; it corresponds approximately

to the logarithmic biomass of organisms in the size range of herbivorous crustaceans[58]. $A \geq 0$ is the amplitude of modulations. For perfectly sinusoidal dome patterns, $10^{2A}$ would be the ratio between the biomasses of organisms in the size classes at the peak of domes and on the bottom of neighbouring troughs. Because of deviations from the sinusoidal form, however, the actual ratio tends to be larger. The parameter $D \geq 0$ quantifies the separation of domes on the one $\log_{10}$ body mass axis; the body mass ratio of organisms occupying neighbouring domes is $10^D$. The parameter $P$ controls the phase of modulations. It shifts the position of domes along the size axis. Below we do not consider it, because its direct comparison across data sets is difficult when $D$ is not fixed. To avoid overparameterization, Eq. (1) does not consider the theoretical possibility of increases or decreases of modulation amplitude along the size axis.

Both empirical and model size spectra occasionally contain gaps, i.e., body size ranges without detectable biomass[62–64] (Fig. 1a, b). We used non-linear median regression to fit Eq. (1), which allowed us to represent these gaps by arbitrary, numerically small biomass densities without biasing the fits. In Supplementary Note 4, we present graphs of all 25 data sets and corresponding fits.

Our data comprise a wide range of ecosystems differing in respect to latitude, size and depth, which affect phenomena not modelled in the SSSM, such as prey defence and allochthonous inputs. To validate the SSSM, we therefore do not attempt a quantitative fit to data but constrain ourselves to the objectives of pattern-oriented modelling[65]. That is, we aim to reproduce qualitative patterns in the empirical responses of the size-spectrum characteristics $B_0$, $S$, $A$ and $D$ to enrichment, and the order of magnitude of these effects. As shown in Fig. 3, the SSSM reproduces well the overall increase in system biomass with enrichment[62] (Fig. 3a). Also in good agreement with observations, enrichment in the model leads to an increasing (less negative) size-spectrum slope[32] $S$, with the increase becoming less pronounced at higher nutrient levels (Fig. 3b). Importantly, both data[32] and model follow our theoretical expectation (Fig. 1c) that size-spectrum modulations become stronger with enrichment (Fig. 3c). At low nutrient concentrations and resulting small modulation amplitudes, the separation between domes $D$ is ill-defined for both data and model. Once pronounced dome patterns arise, data and model agree in that the separation between domes $D$ is not much affected by trophic status (Fig. 3d, Supplementary Note 7). Thus, all major patterns in the data are reproduced by the SSSM, validating it as a good description of pelagic size spectra.

**Mechanisms**. As is common in the study of self-organised periodic patterns[42,66], key insights into the mechanisms driving dome formation in size spectra can be gained from studying the linear response of the system to small perturbations. From a previous mathematical analysis[25], it is known that pelagic size spectra exhibit a superposition of three mathematically distinct linear responses to pressure on a single body size class (Fig. 4, Methods, Supplementary Note 6): (i) the top-down cascade, modulating the size spectrum towards lower body size classes[22,24], i.e., leading to alternating enhancement and depletion of biomass along the size axis; (ii) a modulated bottom-up cascade[21,24,30] and (iii) the conventional, unmodulated bottom-up effect, consistently either enhancing or depleting the biomasses of all larger-bodied species. Depending on system parameters, the two modulated responses can be either amplifying or attenuating (in terms of proportional changes in abundance) as they propagate away from the pressure along the size axis[25] (Fig. 4). The conventional bottom-up effect always increases as it travels up the size axis[25], a phenomenon called trophic amplification[67,68].

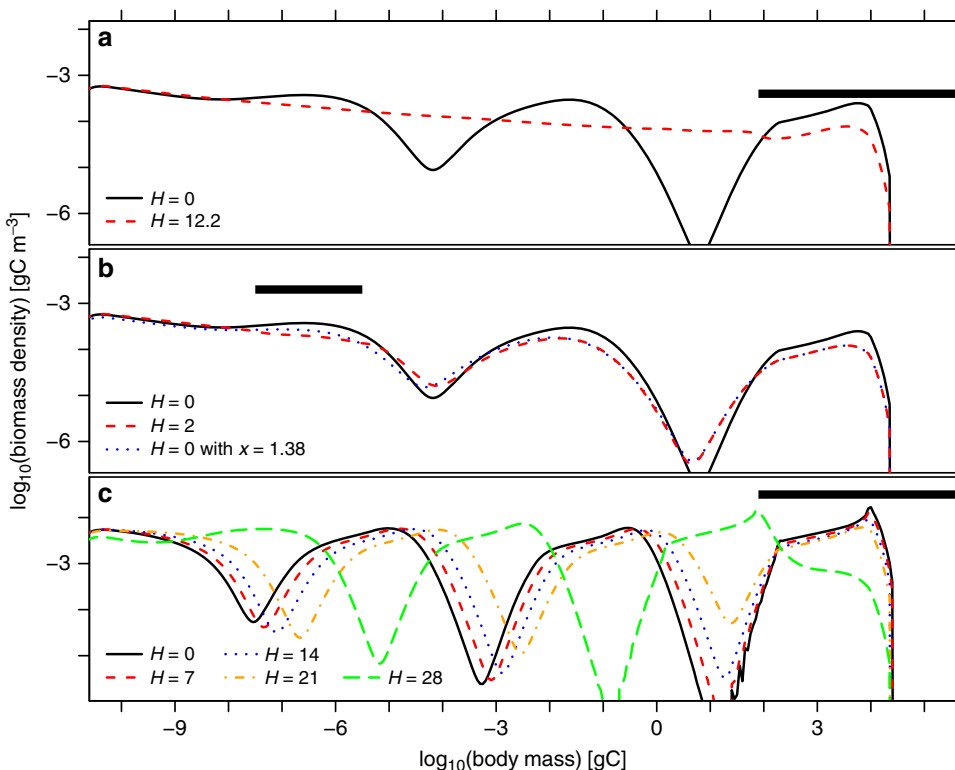

**Fig. 2** Demonstration of top-down and bottom-up effects in SSSM simulations. Species with body mass at first maturation $m_*$ in the ranges indicated by the horizontal bars (panels **a**, **c** $m_* \geq 10^{-1.9}$ gC; panel **b** $10^{-7.5}$ gC $\leq m_* \leq 10^{-5.5}$ gC) are harvested at an allometrically scaled rate $(m_*/\text{gC})^{-1/4}H$ with $H$ as given in the legend. Panels **a**, **b** show community size spectra for moderate nutrient supply (eutrophication parameter $x = 10^{0.2} = 1.58$). The dome structure can be top-down controlled (**a**) but not bottom-up controlled (**b**). In panel **c**, nutrient supply is higher ($x = 10$) and dome size controlled through inherent non-linear regulation[72] rather than cascading pressures. Contrasting panel **a**, top-down forcing $H$ in panel **c** therefore barely affects the height of the size-spectrum mode near $10^{-5}$ gC body mass; only the position of the mode shifts

Dome formation in the SSSM corresponds to a transition from attenuating to amplifying top-down cascades with increasing enrichment, as we demonstrate in a mathematical analysis of the mechanisms at work in Supplementary Note 7. The analysis identifies two distinct mechanisms driving this transition. The first depends on the fact that, due to trophic amplification, the size spectrum slope $S$ increases with enrichment (Fig. 3b). Because consumers tend to be larger than their resources, this increases the abundance of consumers relative to their resources in enriched systems. As a result, a given proportional change in consumer abundance can affect a larger proportional change in resource abundances. The second mechanism is driven by the overall increase of community biomass in enriched systems. As detailed in S7, this leads to partial satiation of consumers and reduces their ability to control their resources, because (i) their food intake rate becomes less dependent on resource abundance and (ii) resources of satiated consumers experiences a safety-in-numbers effect[69]. Both enhance top-down cascades, as is the case in simple mathematical models of infinite[13] and finite[4] food chains.

It should be noted that the 'top-down' and 'bottom-up' terminology above refers only to the direction of propagation of effects in size spectra. At the food-web level, the underlying processes can be more complicated[25,70,71]. This may explain why, despite providing some indications for top-down cascades[70,71], empirical studies of feeding interactions in size spectra did not fully reveal the nature of dome formation.

Generic arguments developed in the study of self-organised periodic modulation patterns in physics[42,66] imply that, for system parameters far beyond the onset of modulations, the phase of modulation patterns still easily responds to pressures, but not the amplitude. Instead, the modulation amplitude is controlled by inherent non-linear regulation[66,72]. For size spectra, this means that harvesting biomass from one dome does not affect the height of other domes (relative to neighbouring troughs), only their positions along the log(body mass) axis. This could explain why experimental manipulations of the top trophic levels in pelagic communities sometimes do not comply with predictions from simple food-chain theory[2]; often producing pronounced changes in mean zooplankton body size rather than in total zooplankton biomass[2,3]. We demonstrate this effect in Fig. 2c. This sensitivity of dome positions to pressures might also lead to a tendency for domes to form in biologically favoured size ranges, for which there is some empirical evidence[20].

**A mechanistic explanation.** Above, we have explained dome formation as a consequence of nutrient enrichment in three complementary ways[73]: first, by obtaining the empirical relation between dome amplitude and nutrient concentration directly from observation data (Fig. 1a; black circles in Fig. 3c); second, by simulating a process-based model, which demonstrates that individual-level processes lead to dome formation with increasing primary production (Fig. 1b; red crosses in Fig. 3c) and third, through a mathematical analysis of this model, which reveals step-by-step the causal chain leading from enhanced primary production to the onset of dome formation in a transition from attenuating to amplifying top-down cascades (Fig. 1c; Supplementary Note 7). The present work thus adds to the small number of cases in community ecology where an analytically

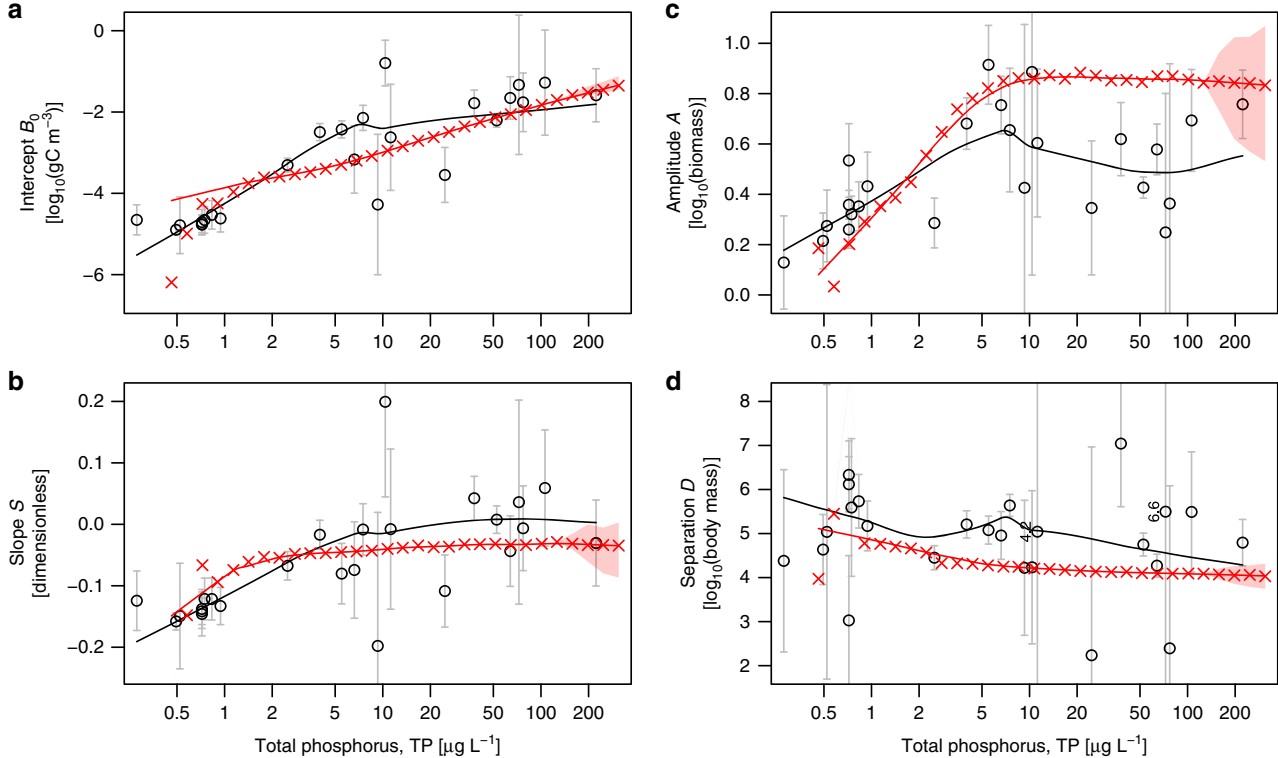

**Fig. 3** Dependence of community size structure on nutrient enrichment. The four panels quantify intercept $B_O$ (at 1 µgC) and slope $S$ of size spectra, the amplitude $A$ of domes and their separation $D$ on the $\log_{10}$-body mass axis, as defined through Eq. (1). Empirical data (black circles, ±s.e.) and medians over 100 simulation snapshots (red crosses, 1st to 3rd quartile ranges are shaded in pink when the model exhibits steady-state dynamics, see S3 for details). Model runs are for eutrophication parameter $x = 10^{-0.1}, 10^{0.0}, 10^{0.1}, \dots, 10^{2.8}$, assuming chlorophyll-$a$ concentrations $0.2x$ µg L$^{-1}$ and converting to TP (Methods). Lines are weighted LOESS smothers. Weighted linear regressions (Supplementary Note 5) reveal statistically significant ($p \le 0.01$, two-sided) dependencies on $\log(\mathrm{TP})$ for empirical intercepts $B_O$, slopes $S$ and dome amplitudes $A$, and a weak but statistically significant ($p = 0.042$) decreases of dome separation $D$ with $\log(\mathrm{TP})$. In cases where error bars extend across graphs, numerical s.e. are written next to data points, thus indicating high ambiguity of the underlying data. Because we weighted empirical data according to inverse squared s.e. in all analyses, these high s.e. data sets contribute less to analyses than, e.g., the highly consistent and accurate measurements for Lake Superior (TP = 2.5) or Lake Constance (TP = 52.5)

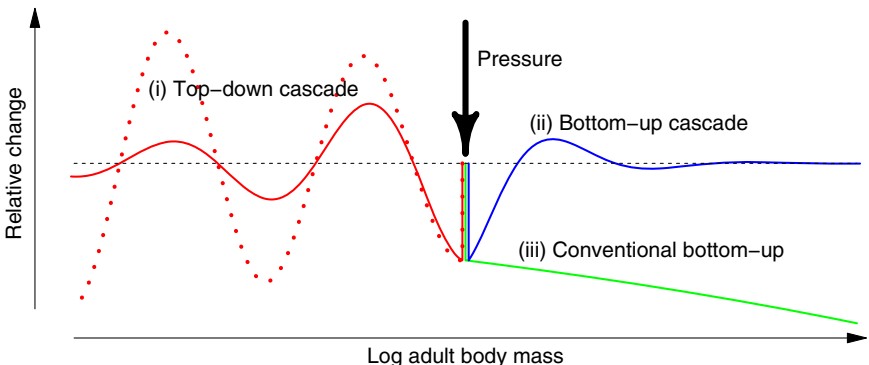

**Fig. 4** The three main distinct responses of size spectra to size-specific pressures. When species of a specific body-size class are continuously removed from the community (thick arrow), theory predicts (i) a top-down modulation of population abundances towards smaller body-size classes, which may be either attenuating (solid red line) or amplifying (dotted red line); (ii) a bottom-up cascade in the reverse direction (blue) and (iii) an amplifying conventional bottom-up effect (green). Depending on the size class at which the pressure is applied, not all of these effects may unfold. The horizontal dashed line represents the pressure-free state. We find that domes emerge when the top-down cascade becomes amplifying towards smaller body sizes

tractable, process-based model semi-quantitatively explains a rich empirical phenomenology[74–76]. This became possible not only by the judicious construction of our model from simple components (Supplementary Note 1), but also by developing mathematical methods that allowed us to analyse the model despite its complexity (Methods, Supplementary Notes 6 and 7).

The mechanistic analysis permits us to address questions regarding the generality of the process of dome formation. For example, one might ask to what extent dome formation is affected by life-history parameters such as the relative sizes of new offspring, individuals at first maturation, and the largest adults of a species—especially since these proportions vary considerably

among species and along the size gradient. Our analysis suggests this dependence is weak: these proportions do not enter the expressions that control the transition to dome formation (Supplementary Note 7). The approximate analytic expression that we derive for the distance $D$ between domes along the logarithmic body mass axis [Eq. (29) of Supplementary Note 7] depends only on two parameters characterising the typical range of body mass ratios between individual predators and their prey, and an allometric scaling exponent.

Another implication of the mechanistic theory follows from the fact that it makes heavy use of the assumption that predators feed on living prey smaller than themselves. The mechanisms identified are unlikely to operate effectively in food webs that are not as strongly size structured as pelagic communities. In benthic communities size structure is less pronounced[77]. One can therefore plausibly expect that the dome patterns found in benthic size spectra[78,79] are controlled by different kinds of mechanisms[79,80], and thus respond differently to enrichment than those found in pelagic communities.

Our explanation of dome formation as a transition from attenuating to amplifying top-down cascades is consistent with the conclusion of an earlier meta-analysis that top-down cascades tend to be attenuating in marine pelagic systems but have no such general tendency in the nutrient richer lake pelagic communities[6]. As explained above, the general theory of pattern formation suggests that far beyond the onset of dome formation the dome amplitude is controlled by inherent non-linear regulation[66,72], rather than by the amplitudes of neighbouring domes. When top-down cascades are amplifying in a linear model, as we find for nutrient rich pelagic systems, non-linear effects thus constrain the maximum height that domes eventually attain[72]. This explains why, on average, trophic cascades observed in lake pelagic communities are neither amplifying nor attenuating[6].

Our analytic calculations (Supplementary Note 7) constrain the transition point from attenuating to amplifying cascades to the range $1 < x < 8$ for the model's eutrophication parameter, corresponding to $0.6\,\mu g\,L^{-1} < TP < 4.4\,\mu g\,L^{-1}$ or chlorophyll-$a$ concentrations in the range $0.2$–$1.6\,\mu g\,L^{-1}$, with lower values preferred when size-spectra span longer body-size ranges. Coastal marine waters often lie in this transition range (which is below levels where harmful algal blooms tend to occur[81,82]). On this basis, we predict the occurrence of dome patterns in nutrient rich coastal marine waters. Climate warming can increase coastal surface nutrient concentrations further[68], thus intensifying this effect, with potential implications for marine ecosystem management. However, we caution that the transition from attenuating to amplifying top-down cascades does not result in a sharp qualitative transition in observed size spectra. For example, the cascade in the unperturbed model size spectrum for $x = 1.58$ (Fig. 2a, black line) is still attenuating (compared to neighbouring domes, the right trough is deeper than the left), yet it already exhibits a clear dome structure.

The analytic calculations also reveal why models simpler than the SSSM would struggle to convincingly explain dome formation. All details of the SSSM that distinguish it from previous, simpler size-spectrum models have a role to play in reproducing and explaining the phenomenology we report. Size-spectrum models that do not distinguish individuals by maturation body mass[22,23,45] are unable to represent in their steady state the gaps observed in size spectra[62,63], because there is no biological mechanism to generate individuals of body sizes larger than the size class of a gap. Models without any representation of intraspecific size structure[31] would, amongst others[25], overestimate the magnitude of domes and the prevalence of gaps. Models representing communities as food chains[13] rather than as continua of species cannot represent shifts in the positions of

domes in response to pressures[66], as seen in Fig. 2c. Models without representations of consumer satiation[22,23] omit a model component that is essential for the transition to amplifying cascades. Models that include non-linear stock-recruitment relationships in addition to[24] (rather than implied by[50]) density-dependent feeding interactions are unlikely to generate amplifying top-down cascades or bottom-up amplification, because they tend to overestimate the strength of inherent regulation of population size. Models that explicitly evaluate food webs of size-structured populations without imposing artificial[16] limits to recruitment[48,50], i.e., the type of models that the SSSM approximates, tend to be computationally too expensive for simulations of species-rich communities over size ranges as large as studied here. The SSSM's demonstrated ability to describe high-level phenomena in pelagic ecosystems over a wide range of conditions suggests future applications in diverse areas such as the testing of ecological indicators, development of management strategies or ecological forecasting.

The significance of our results lies not only in the characterisation of dome patterns or top-down cascades, but crucially in the identifications of the former as manifestations of the latter. This became possible by understanding dome formation as part of a wider scenario of community change with increasing enrichment (Fig. 3). In particular, the results speak against the widely cited idea that domes represent bottom-up cascades[21,30,31]. While bottom-up cascades have indeed been demonstrated in the SSSM and similar models[24,25], our mathematical analysis (Supplementary Notes 6 and 7) shows that these cascades get damped by increasing enrichment, opposite to the observed pattern (Fig. 3). Based on the classical theory, we had also expected that the flattening of the dome with smallest body size in model simulations (Fig. 2b) would similarly reduce modulations along the entire size-spectrum, which was not the case.

**Relation to detailed empirical accounts**. When comparing our theory with detailed observations in specific ecological communities, it must be kept in mind that any size-spectrum model is a simplified high-level description. It relates to low-level descriptions in terms of populations and their interaction networks in an analogous manner as a macroscopic description of sound waves in an air-filled chamber relates to a microscopic description of the air in terms of freely moving and occasionally colliding molecules. Concepts that are central to a high-level description (domes and sound waves) are unnecessary or even meaningless for a full description and explanation of dynamics at the lower level. When a detailed low-level description is available, one can, a posteriori reconstruct the macroscopic phenomenon (modulation of community biomass along the size axis for domes, density waves for sound), but this does not add information to that already provided at the lower level. To the contrary, since high-level descriptions are generally approximations, their juxtaposition with corresponding low-level descriptions will necessarily reveal inaccuracies.

Such juxtapositions, relating the populations and interactions of species and functional groups to the resulting size spectra, are available, for example, for Lake Constance and Lake Müggelsee[58,59,70,83,84]. One particular effect that size-spectrum models cannot capture, which was identified in these studies, are changes in the distributions of traits other than size[85]. Yet, such changes must be expected as part of a re-organisation of community structure, e.g., in response to nutrient enrichment. For example, it has been observed that the size range covered by carnivorous zooplankton may change[58,83] and less edible phytoplankton may pile up in certain size classes of the autotrophic size range, so

reducing the flow of biomass towards larger consumers[83]. Conversely, particularly efficient and competitive consumers such as daphnids may obtain biomasses above the average zooplankton due to their ability to exploit relatively small and thus highly productive prey[58,70].

Such low-level descriptions and the higher degree of detail they provide do not, however, invalidate high-level descriptions, e.g., in terms of size-spectrum models. Low-level descriptions tend to be system specific and would struggle, for example, to provide simple explanations of trends seen across systems, such as those documented in Fig. 3—while the SSSM achieves just this.

**Conclusion**. Along multiple lines of evidence we have demonstrated that the dome patterns found in pelagic size spectra of lakes are likely governed by top-down trophic cascades and moderated by the availability of nutrients. The frequent observation of pronounced dome patterns[20,27,31,63,70,83,86], where variation in biomass often exceeds a factor 100 (Fig. 1a), therefore suggests that strong top-down cascades are common in freshwater communities. These powerful trophic cascades, generating domes, are active in pelagic systems without any manipulation of top trophic levels[2,3]. Pelagic dome patterns might therefore be the clearest cases yet of self-organised ecological pattern formation in trait space.

The results of this study imply that measurement of the strength of top-down cascades in lakes does not necessarily require experimental manipulations or comparative studies. Cascade strength can be estimated directly from the modulation amplitude of size spectra (Fig. 3c). Application of this idea to the old question of how cascade strength depends on nutrients provides clear evidence for an overall increase of cascade strength with increasing nutrient concentration in both model and data (Fig. 3c) up to around 5–10 µg L$^{-1}$ TP. Beyond this level there might be plateau or even a slight decline in cascade strength with TP. Thus, several of the patterns previously considered[9] appear to be combined. To further clarify the details of this dependence, we call for systematic measurements of size spectra across nutrient enrichment gradients, aided by the accurate automated methods now available[27]. Monitoring of coastal marine size spectra might guide the interpretation of ecosystem changes when domes form unexpectedly.

For bio-manipulation of top predators[2,3], we predict that measurements of size spectra in lakes will, depending on trophic status, reveal responses of the amplitude or the phase of the dome pattern (Fig. 2). Responses to pressures on high-ranking

predators will therefore not always follow expectations from simple food-chain theory[2,3], but transfer of pressures to lower trophic levels should generally be expected in the light of the new theory. The general theory of non-linear pattern formation thus provides new mechanistic insights into the structure and dynamics of ecological communities.

## Methods

**Representation of size spectra**. For the purpose of comparison across studies, empirical size spectra are often represented on double-logarithmic axes in the so-called *normalised* form[87,88]: the volume (or areal) density of biomass of organisms measured in each body mass interval considered is divided by the linear width of this interval (of dimension Mass). In the limit in which the width of these intervals goes to zero, this represents the density of a community's biomass (per unit volume) along the linear body mass axis[89]. A disadvantage of this representation is that normalised spectra tend to spread over a wide numerical ranges. Superimposed "dome" modulations are not easily visible.

To overcome this disadvantage while maintaining comparability of spectra across studies, we computed not the density of biomass (per unit volume) along a linear but along a logarithmic body mass axis. Specifically, the natural logarithm of body mass was used, which permits a simple conversion of traditional normalised size spectra to this density-along-the-log-axis form: one just needs to multiply each normalised size spectrum value by the corresponding body mass. To obtain unbiased estimates of the continuous size spectrum from empirical data for discrete size intervals one multiplies with the geometric mean of upper and lower interval boundary[25]. This is the representation used throughout the present study:

$$\text{size spectrum at point } m_i = \frac{B_i m_i}{\Delta m_i}, \tag{2}$$

with $m_i$ denoting the geometric mean of the boundaries of body mass interval $i$ (i.e., the mid point on a log axis), $\Delta m_i$ its linear width, and $B_i$ the measured biomass volume density of individuals with body mass lying in this interval. Equation (2) is formally equivalent to a known heuristic "denormalisation" procedure for size spectra[33,87,88].

**Data sets**. Searching for the keywords "size spectrum" and "size-spectrum" in literature databases and following relevant citations, we identified 25 pelagic size-spectrum datasets that satisfied our inclusion criteria (good technical quality, coverage of a 10$^6$ body mass range, and quantification of trophic status). Except for a data sets from Lake Malawi, all sampling locations lie within the latitudinal band 28°–54°N (Fig. 5).

In some cases, uneven sensitivity of sampling and use of different sampling methods over different size ranges can lead to structures in the data that resemble dome patterns. Occurrence of such artefacts, however, can be recognised by comparing method boundaries along the size axis and dips or discontinuities in spectra, and usually such issues are acknowledged by study authors. We excluded two studies because of such concerns[90,91]. While uneven sampling might also have contributed to some of the unevenness in other spectra (just as any empirical method has potential biases), the fact that we find a clear signal of increasing dome amplitude $A$ with increasing nutrient richness (Fig. 3) in accordance with earlier observations[32] would be hard to explain if sampling artefacts where generally the dominating contribution to size-spectrum modulations.

We converted chlorophyll-*a* concentrations [Chl-*a*] to total phosphorus (TP) concentrations TP using the relation[92] ["This study (all lakes)" in the source]

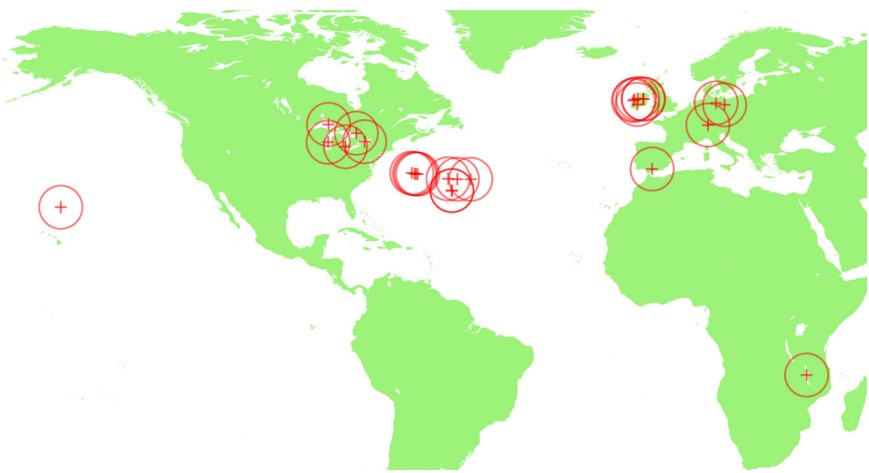

**Fig. 5** Locations of study sites included in analysis

$\log_{10}[\text{Chl-a}] = -0.455 + 1.026 \log_{10}(\text{TP})$, i.e., $\log_{10}(\text{TP}) = (\log_{10}[\text{Chl-a}] + 0.455)/1.026$, with [Chl-*a*] and TP given in $\mu g \, L^{-1}$. The conversion was used mostly for marine systems, where nutrients other than phosphorus may be (co-)limiting. The TP values so obtained have therefore purely nominal character as expressions of overall nutrient availability.

The following lists the systems underlying the empirical data in Figs. 1 and 3, together with data sources for size spectra and, if different, sources for TP or [Chl-*a*] (expressed in $\mu g \, L^{-1}$): North Pacific Central Gyre[33] with[93] [Chl-*a*] = 0.0945 (corresponding to TP = 0.28); stations Purple 10 ([Chl-*a*] = 0.17, TP = 0.49), Purple 11 ([Chl-*a*] = 0.18, TP = 0.52), Yakutat ([Chl-*a*] = 0.25, TP = 0.72), Nashville ([Chl-*a*] = 0.25, TP = 0.72) and Indigo ([Chl-*a*] = 0.26, TP = 0.75) in the New England Seamounts Area[94] and stations Sargasso 12 ([Chl-*a*] = 0.25, TP = 0.72), Sargasso 14 ([Chl-*a*] = 0.29, TP = 0.83) and Sargasso 13 ([Chl-*a*] = 0.33, TP = 0.94) in the Sargasso Sea[94], where data points below $5 \cdot 10^{-13}$ gC body mass were discarded due to acknowledged methodological artefacts[94]; Lake Superior[27] with[95] TP = 2.5 (average of years 2006 and 2011, numerical data by Peder M. Yurista, priv. comm.); the averaged spectrum of 37 inland lakes in central Ontario[20,62] with median TP = 4[20,62]; Lake Michigan[96] with TP = 5.5[97]; the six Irish lakes[98] Loughs Maumwee (TP = 6.6), Carra (TP = 11.2), Gara (TP = 24.7), Gur (TP = 37.9), Mullagh (TP = 72.6) and Ramor (TP = 77.1, numerical data by Elvira de Eyto, priv. comm.); Lake Ontario[86] with TP = 7.5[95]; Lake Malawi[86] with TP = 9.3[99]; Lake St. Clair[32] (average over 14 stations, TP = 10.4); Lake Constance[58] averaged over years 1987–1996 (unpublished data by U.G.) with TP = 52.5[100]; Fuente de Piedra[101] ([Chl-*a*] = 25[102], TP = 64.0); Arendsee[63] (TP = 106); and Müggelsee[83] averaged over years 1988–1990 (numerical data by U.G., TP = 223). Size-spectrum data were extracted from referenced published graphs if not stated otherwise. When publications contained multiple graphs with size spectra for a given system, we selected the spectra averaging over the longest time interval and the largest number of stations. CSV files of the size-spectra analysed in this study are included as Supplementary Data 1.

**Missing empirical values**. Size-spectrum values of zero are sometimes suppressed in empirical data. In published size-spectrum data sets, we therefore identified as a 'gap' any occurrence of a large interval between subsequent reported size-class midpoints. Precisely, any spacing on the log-body mass axis between subsequent reported size-class midpoints that was over 1.8 times wider than both the previous and the subsequent spacing was considered a gap. Each such gap was filled by a single size-spectrum value of zero at the centre of the gap. All reported values of zero were retained, except for those at the upper end of reported body mass ranges, since these might be due to insufficient sample volumes. Before taking logarithms, size spectrum values of zero were replaced by $10^{-100}$ times the smallest reported non-zero value. The thus processed size spectra are included in Supplementary Data 1 as an R object.

**Size range used to determine characteristics**. For most empirical data sets (84%), the body size range covered does not extend beyond 0.1gC. For comparability across empirical data sets and between data and simulations, only size-spectrum data up to 0.1 gC were therefore used when fitting Eq. (1) to extract the characteristics $B_0$, $S$, $A$ and $D$. We stress that this restriction of the size range is a decision about how to characterise the size spectra. It does not imply a statement about what size range is dynamically relevant in either models or reality. The role of the distinction between models and characterisations in the development of ecological theory is discussed in ref. [13].

**Non-linear median regression**. To fit Eq. (1) to size spectrum data, we used the function `nlrq` from the package `quantreg` (v. 5.36) of the R programming language, v. 3.5.3[103], which implements an interior point method for non-linear quantile regression[104]. The function requires initial parameter values for a non-linear optimisation routine, which we set to $B_0 = -2.5$, $S = -0.1$, $A = 1$, and all combinations of $D = 4, 6$ and $P = 0, 0.4\pi, 0.8\pi, 1.2\pi, 1.6\pi$. When different initial values resulted in different fits, the result best satisfying the quantitative optimisation criterion of the fitting method was selected. However, values for dome separation were constrained to the range $D = 2–9$, because size spectrum modulations with very small or very large wavelengths were not phenomena of interest to the present analysis. Standard error estimates for the empirical fitting parameters were computed using the jackknife method.

**LOESS smoothes**. Smoothes in Fig. 3 are first-order LOESS based on the closest 2/3 of data points (1/3 for simulations), weighted by tricubic distance and for empirical data in addition by inverse squared estimated standard error. To improve the robustness of the smoothes, they were computed using M-estimation with Tukey's biweight using the function `loess` of R.

**Mathematical analysis**. We outline the mathematical methods used to establish the mechanisms driving the transition from attenuating to amplifying top-down cascades in the SSSM. For details see Supplementary Notes 6 and 7.

The standard methods[42] to identify the conditions and driving mechanisms for the formation of self-organised periodic patterns, i.e., spatial modulation of some system property, relies on the computation of the linear growth rate (in

time) of small, sinusoidal perturbations of the unmodulated base state of the study system. When modulations grow through time, patterns form. But this method is not applicable here. Because of allometric scaling of physiological rates, the dynamics of small species tend to be much faster than that of large species, so that periodic modulations do not have well-defined linear growth rates (they are not eigenfunctions of the linearised dynamics). The method used here[25], therefore, considers instead the static, equilibrium linear response of the system to small, sustained and localised press perturbations. For this, the allometric scaling plays no essential role. Patterns form when the equilibrium response increases with the distance from the perturbation along the size axis. In Supplementary Note 6, we demonstrate for an exactly solvable example that in cases where both methods are applicable they give equivalent results.

In size-spectrum models, the equilibrium response to localised pressures can be decomposed into a sum of sinusoidal responses that, starting from the perturbed size class, grow or decline exponentially along the logarithmic size axis, plus unmodulated exponentially growing or declining components (as in Fig. 4). In addition to these exponential components, the static response may have a localised core residual that declines faster than exponential with the distance from the perturbation[25]. Each component of the sum has a characteristic complex-valued wave number, the real part of which is $2\pi$ divided by the wavelength of the sinusoidal modulation, and the imaginary part the rate of exponential growth or decay along the logarithmic size axis.

To determine these wave numbers, one first needs to compute the effective interaction kernel for species along the size spectrum. This function describes the dynamic linear response of species in all size classes to changes in the abundance of species in one given size class. The interaction kernel thus encapsulates the underlying ecology. The wave numbers of the linear modes of the size spectrum are give by the zeros, in the complex plane, of the analytic continuation of the Fourier transform of this interaction kernel.

While for the SSSM this Fourier transform is a rather complicated mathematical expression, it turns out that at most points in the complex plane just a few terms of this expression (associated with specific ecological phenomena) dominate numerically. Hence, the locations of the zeros can be understood from the properties of just a few terms.

In our mathematical analysis, changes in the interaction kernel due to nutrient enrichment are described by a heuristic modification of the kernel that corresponds to an overall increase in food availability. Using this to study the effects of enrichment on the Fourier transform of the interaction kernel, the mechanisms driving the transition from attenuating to amplifying top-down cascades are then identified.

**Reporting summary**. Further information on research design is available in the Nature Research Reporting Summary linked to this article.

## Data availability

All data analysed in this study are included in a Supplementary information file.

## Code availability

The simulation code used in this study is included in a Supplementary information file.

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

## Acknowledgements

We thank Elvira de Eyto and Peder M. Yurista for making numerical size-spectrum datasets available for this study, and Stephen R. Carpenter, W. Gary Sprules, Jonathan B. Shurin and Mark Trimmer for comments on earlier drafts of this paper. A.G.R. was funded by the Natural Environment Research Council and the UK Department for Food, Environment and Rural Affairs (Defra) within the Marine Ecosystems Research Programme (MERP, NE/L00299X/1).

## Author contributions

A.G.R. developed the non-linear SSSM and performed its mathematical analysis. A.G.R., U.G. and P.K. collected and critically evaluated the literature data and developed the methods for model–data comparison. U.G. contributed unpublished data sets. All authors jointly wrote and edited the paper.

## Competing interests

The authors declare no competing interests.
