## [Peer Review File · Nature Communications]

Reviewers' Comments:

Reviewer #1:

Remarks to the Author:

This paper tackles an important topic, integrating life history, predation, food web structure and clustering of species by trait. The manuscript compares predictions of a mathematical model (developed primarily in previous works) to empirical data drawn from a survey of food webs with size-structure data. My comments follow.

** Mathematical Model.

— Overall, there is not a sufficient description of the mathematical model in either the main paper or the supplementary information. I understand there are limits to what can be included in the former, but I think at least the defining equations, perhaps along a conceptual figure showing the schematics of the model, would be helpful to orient readers. In the latter, the authors several times refer readers to previous work, but I think there needs to be more. The analysis here should come closer to standing alone.

— Perhaps in part because there isn't enough information in the SI, I was confused from the beginning as to what defines the effective growth rate for size class m^* . This seems to be clearly defined in Eq S1. But then another term is added in Eq S3, seemingly for pragmatic reasons, but I could also reinterpret S3 as simply redefining the growth rate $\Lambda(m^*)$, since it is linear in $B(m^*)$. How is this additional term to be interpreted ecologically/mechanistically? I don't quite understand whether/how it is in the model used to calculate $\Lambda(m^*)$ in the first place, and if it isn't, how we justify modifying one of the central equations in this way.

— The scheme for computing responses to press perturbations in Supplementary Sections S6 and S7 should be referred to more explicitly in the main text, as this is really the critical part of the analysis. I would suggest demonstrating the approach on a cut-down simplified version of the authors model, perhaps even a case (however unrealistic) where the responses to press perturbations can be computed analytically. This will give readers a better intuition for the overall approach, and for the approximations used e.g. in considering only the first few terms here.

** Conceptual advances.

— Many places in the text compare this approach/results to more general ideas of pattern formation across biology and elsewhere. Sure—there are analogs. But in my reading this is (a) too prevalent in the text. These examples do not add much and the current analysis basically stands alone. (b) not dealt with carefully enough where it could be useful. For example, the standard way (that at least I am familiar with) to think about Turing instabilities is as a result to small perturbations which are then allowed to relax. These take an unstable equilibrium solution towards (or not) a patterned, periodic solution, and the analysis proceeds by considering the eigenvalues of various modes, the sign of their real parts, etc.

This is distinct from the press perturbations considered here, where one or more species abundances is essentially permanently reduced (or increased). This more likely corresponds to a case where a parameter in the model (like harvesting rate, as mentioned by the authors) is perturbed, as opposed to where abundances are instantaneously perturbed as in the case above. But these distinctions are glossed over to my mind, making the comparison with other pattern-forming systems confusing at best.

— I would like to see a clearer set of links to other, existing work on top-down and bottom-up pattern formation in this context. The paper cites other work adequately, but I was left a bit confused about what would have to be omitted from this model to end up with (for example) only the possibility of top down cascades. In what limiting cases would the authors say their model reduces to other approaches? I also note a very recent paper which qualitatively reaches some similar conclusions (Barbier and Loreau, Ecology Letters, 2018)

** Comparison with data

— I was not totally sure how to evaluate how well the model performs against these data sets, but I do respect the effort the authors have gone to to assemble these data sets carefully and then perform the comparisons. Where I was stuck was this: the authors make quite a big deal of the fact that their approach can predict much more than highly simplified models, which might (say) predict domes, but nothing more precise than that. But at the same time, the authors' own comparison with the data was only semi-quantitative, and I wasn't sure what would count as a poor vs a good fit. Basically—readers should understand how well the model did, beyond just eyeballing the fits.

This is compounded by the fact that the model actually fitted to the data (Eq 1 in the main text) is if I understood correctly itself a phenomenological fit from simulated data. I.e. not actually an analytical outcome of the model, but instead an approximation that works well (in some? all?) parameter regimes. So if this phenomenological fit fails, does that mean that the underlying, full model fails? Or could Eq 1 sometimes fail to fit both empirical and simulated data in the same ways. This could be made clearer.

Reviewer #2:

Remarks to the Author:

Review of manuscript: "Dome patterns in pelagic size spectra reveal 2 strong trophic cascades"

Overall comments

Using empirical data, simulations and the existing framework of Species Size Spectrum theory the authors offer a novel interpretation on observed domes in biomass formed across a wide size spectrum. Existing literature suggests that domes form as a result of a bottom up effect whereby each subsequent dome in biomass of a specific size spectrum arises as a result of the dome on its left side. The authors reconstruct the framework suggesting the opposite: that it is the higher predators that essentially drive the dome formation through top down effects, and this effect is pronounced under eutrophic conditions. Within this framework, apparent contradictions within food chain as well as size spectrum theories seem to be resolved.

I can see this approach having its merits for ecosystem modelling which typically tends to over simplify a system by focusing only on its emergent properties of eg energy flow and nutrient cycling by averaging out important interactions between species populations and traits. This essentially "black box" modelling approach often compromises flexibility (i.e. maintaining variability in space and time as conditions change). This manuscript might offer a way out: Knowledge on the size domes represented in a specific ecosystem and the reasons leading to the emergence of these domes could in the future replace the rigid "black box approach" in representing biota, still without overcomplicating models by including foodweb interactions and their strengths.

I believe the idea presented here is important and is offering a refreshing new perspective for ecological theory underlying foodwebs. The figures offer useful conceptualisation for understanding the patterns and mechanisms. Having said that, I believe that certain points need improvement and specifically:

1. The introduction must be more focused avoiding redundant information

2. A description and ecological implication of the size spectrum characteristics should be provided
3. The mechanisms must be explained in a more detailed manner
4. A more careful discussion on the implications of the relevance of this framework for coastal ecosystems should be attempted.

I also found that the inclusion of lumpy coexistence theory in the introduction was unnecessary for reasons I explain below.

Specific comments

Summary

Line 18: replace by: "Compiling size spectrum data of high-quality..."

Lines 22-25: this sentence is very confusing as it does not follow from the description of the study remit as given above but rather seems to convey an altogether different message that is confusing for the reader.

Line 71: not all of these models are abstract though. Some have an empirical basis by relying on experimental and field data for parameterisation. Thus these sentences must be re-thought.

Introduction

The first two paragraphs need to be more focused and less abstract, connecting better with the summary and aims. Avoid being unnecessarily wordy and connect the two paragraphs into one with a stronger opening line and a punchy last sentence indicating the challenge.

Lines 26-28: the opening statement is too vague...what is implied by "concepts" in this context?

Theories perhaps, frameworks?

Lines 28-29: My understanding was that a food chain consists of only one organism per trophic level as opposed to food webs which represent groups of organisms per trophic level, or else an ensemble of many food chains...

Line 30: "where pressures on top predators", do you actually mean, "pressures of top predators"?

Line 32: Their strengths vary, instead of varies

Line 46: This paragraph is badly connected to the paragraphs above: (1) an alternative perspective to what? To the food chain structure mentioned in the first paragraph? Need to stress the connection. (2) Also please explain why there needs to be an alternative in the first place...eg in order to elucidate, avoid the aforementioned discrepancies..., (3) in which way does this theory help overcome the problems raised above? Be more explicit

Line 56: "For the left-most dome, other explanations are required"...such as? Nutrient control? Offer an example

Line 66-73: I'm really not sure how this paragraph is relevant to the rest of the paper. This paragraph discusses theory on lumpy co-existence which has been supported by mathematical theory (eg Turing's work), abstract consumer resource models, and resource competition models parameterised on experimentally established species traits. Therefore, although the authors are correct in saying that there are no substantial empirical evidence to prove this theory (although limited evidence is presented within Scheffer & Van Ness PNAS 2006), their claim that all models used so far are abstract is incorrect. But more importantly, this theory is referring to species that are competing for similar resources (i.e. species of the same assemblage) and troughs in trait space arise a result of competitive exclusion. Thus, I'm not sure how this conceptual framework can be generalised to describe different trophic levels along a trait space. Moreover, the lumpy coexistence framework suggests that the y axis represents number of species within a certain size class that are favoured as a result of evolutionary processes or long term ecological processes acting on the assemblage. This is conceptually different to the size spectrum theory which suggests that specific size ranges develop more biomass, which, to my understanding, does not preclude the existence of rare species in the troughs.

Line 93: instead of "mechanical" do you mean "mechanistic"?

MODEL STRUCTURE AND BASIC BEHAVIOUR

Line 94: it is not clear what is meant by "abstract but complete description of life histories"

100-108: this naturally leads to the question of what would happen if a species from a middle dome was removed...

Figure 2: "dome size is controlled through internal mechanisms rather than cascading pressures." Not clear what is meant by internal mechanisms here

Paragraph starting line 121: although the size spectrum characteristics are mentioned here, a description of what they actually represent in a biological community should be provided. This would also help readers better understand the mechanisms affecting the patterns across the size spectrum when eutrophication level changes.

MECHANISMS

I believe that figure 3 should become more self-explanatory by including a short description of each characteristic of size spectra.

159-161: it is unclear how the increase in the slope of the size spectrum would lead to an increase of abundance of consumers relative to resources. This needs more explanation to make it explicit

167: Both enhance, instead of enhances

156-168: I find this paragraph rather problematic in describing the mechanism behind the effect of increased eutrophication on the troughs and domes. More care should be given to explain exactly what changes in community structure lead to the troughs.

Discussion

line 216: rephrase

lines 222-223: These considerations

line 220: "nonlinear mechanisms inherent to the system", I'm very intrigued by this phrase, please elaborate

Conclusions:

Lines 270-282: Most of this paragraph is philosophical and redundant. I would omit/merge with next one.

283-293: since we are limiting this to the prediction of community structure in eutrophic coastal ecosystems, we should also consider that most often eutrophication lead to HABs which selectively decrease the abundance of organisms up to the top predator level.

Reviewer #3:

Remarks to the Author:

Dome patterns in pelagic size spectra reveal strong trophic cascades

Axel G. Rossberg, Ursula Gaedke and Pavel Kratina

In aquatic ecosystems, the living biomass as a function of organisms' size is distributed according to quasi-linear log-log size-spectra "from bacteria to whales". A common and well-known feature of these size-spectra is their linearity. However, in some specific ecosystems, and this has been particularly observed in lakes, size-spectra can exhibit distinct stationary domes (e.g. Yurista et al., 2014). In this paper, Rossberg and co-authors attempt to explain these domes using a size-structured community model where an infinite number of species distinguished by their size at first reproduction experience size-based trophic interactions.

The mathematical model developed and used is indeed displaying periodic bumps of increasing amplitude when nutrient enrichment increases (Fig. 1). It is shown that these patterns emerge from bottom-up effects when the abundance of top predators increases (Fig. 2) due to nutrient enrichment (Fig. 3).

While the model proposed is undoubtedly very sophisticated from a mathematical and numerical point of view and thus very interesting for applied mathematicians and theoretical ecologists, I have not been convinced by the ecological conclusions inferred from its analysis and simulations and the discussion of these results. The major issues I had when reading this paper are listed below:

- The domes observed empirically have been shown to match well-identified taxa and functional communities (e.g. viruses, heterotrophic bacteria, unicellular phototrophs, unicellular mixotrophs and heterotrophs, planktonic crustaceans, fish). Each taxa/functional community occupies well-defined and successive size ranges over 3-4 orders of magnitude (e.g. Boudreau and Dickie, 1992; Sprules and Goyke, 1994; Thiebaut and Dickie, 1993; Yurista et al., 2014) due to different limiting processes occurring in these size ranges (Andersen et al., 2016). This is the standard paradigm at the moment and Rossberg et al. don't discuss it and don't explain why it should be abandoned and replaced by a model that is not consistent with the empirical fact that the size ranges of dome-specific taxa don't overlap (in the model, birth size is the same for every species so that the species present in a given dome overlap with all the species present in smaller domes).
- The domes observed empirically are stationary: they don't propagate through the size-spectrum. On the contrary, and unless they explicitly include stabilizing density-dependent processes, size-spectrum models are known to be prone to unstable behavior leading to the formation of traveling waves (i.e. non-stationary domes) that are propagating through the spectrum (Datta, 2010; Hartvig et al., 2011; Plank, 2012; Plank and Law, 2012; Maury and Poggiale, 2013). Despite incorporating an artificial stabilizing term (eqs. S3), the same behaviour is mentioned (but never clearly described) for the model presented when $x > 10$ so that only snapshots are shown in the paper (e.g. Fig. 1). This casts doubts on the homology that Rossberg et al claim between the model behaviour (i.e. unstable traveling waves) and the ecological processes responsible for the presence of stationary domes.
- The presentation of the model in the paper and the supplementary material is confusing and insufficient to understand exactly how this model is constructed, simulated and used.

Overall I believe that the claim that this model explains the dome patterns observed has not been unambiguously and convincingly demonstrated in this paper. Furthermore the model presentation is not really understandable as it stands in the paper and sup. mat. and the discussion of the results in an ecological perspective is insufficient (why should we abandon the standard explanation for the domes?). Lastly, the focus of the paper is maybe too narrow for a generic journal as *Nature Communication* and I would therefore recommend re-submission in a more specialized journal after the concerns expressed here have been addressed.

References:

Andersen K. H., T. Berge, R. J. Gonçalves, M. Hartvig, J. Heuschele, S. Hylander, N. S. Jacobsen, C.

- Lindemann, E. A. Martens, A. B. Neuheimer, K. Olsson, A. Palacz, A. E. F. Prowe, J. Sainmont, S. J. Traving, A. W. Visser, N. Wadhwa, and T. Kjørboe, 2016. Characteristic Sizes of Life in the Oceans, from Bacteria to Whales. *Annu. Rev. Mar. Sci.* 2016. 8:217–41.
- Boudreau, P.R., Dickie, L.M., 1992. Biomass spectra of aquatic ecosystems in relation to fisheries yield. *Can. J. Fish. Aquat. Sci.* 49 (8), 1528–1538.
- Datta, S., Delius, G.W., Law, R., 2010. A jump-growth model for predator-prey dynamics: derivation and application to marine ecosystems. *Bull. Math. Biol.* 72 (6), 1361–1382.
- Hartvig, M., Andersen, K.H., Beyer, J.E., 2011. Food web framework for size-structured populations. *J. Theor. Biol.* 272 (1), 113–122.
- Maury, O., Poggiale, J.-C., 2013. From individuals to populations to communities: a dynamic energy budget model of marine ecosystem size-spectrum including life history diversity. *J. Theor. Biol.* 324 (1), 52–71.
- Plank, M., 2012. Effects of predator diet breadth on stability of size spectra. *ANZIAM J.* 53 (0).
- Plank, M.J., Law, R., 2012. Ecological drivers of stability and instability in marine ecosystems. *Theor. Ecol.* 5 (4), 465–480.
- Sprules, W.G., Goyke, A.P., 1994. Size-based structure and production in the pelagia of Lakes Ontario and Michigan. *Can. J. Fish. Aquat. Sci.* 51 (11), 2603–2611.
- Thiebaut, M.L., Dickie, L.M., 1992. Models of aquatic biomass size spectra and the common structure of their solutions. *J. Theor. Biol.* 159 (2), 147–161.
- Yurista P. M., D. L. Yule, M. Balge, J. D. VanAlstine, J. A. Thompson, A. E. Gamble, T. R. Hrabik, J. R. Kelly, J. D. Stockwell, and M. R. Vinson, 2014. A new look at the Lake Superior biomass size spectrum. *Can. J. Fish. Aquat. Sci.* 71: 1324–1333.

Responses to reviewer comments

We are very much in debt to all three reviewers for their critical reading of our manuscript and the supportive, detailed and stimulating feedback they provided.

Since the previous submission, we found a bug in the simulation code that affected the strength of the damping of short wavelength modulations of the size spectrum that is included in the model to describe food-web effects. We corrected this bug and re-ran the simulations. In the course of this re-evaluation, we also increased the lower boundary of the size-range simulated to a value consistent with the typical size of phytoplankton. In order to permit more than two domes to form in simulations, we correspondingly increased the upper boundary of the simulated size range and increased the upper bound of the size range over which model and data are compared. The results differ in details, but not to a degree that would affect our main conclusions. Noteworthy is that fixing the bug has reduced the tendency of the dome patterns to exhibit steady-state dynamics at high nutrient concentrations, which leads to cleaner simulation results.

All other changes to the manuscript are discussed in our point-by-point response to reviewer comments below.

All new or modified text is marked in blue in the revised manuscript. Line numbers in our response to the reviewers below refer to the revised manuscript.

Reviewer #1 (Remarks to the Author):

This paper tackles an important topic, integrating life history, predation, food web structure and clustering of species by trait. The manuscript compares predictions of a mathematical model (developed primarily in previous works) to empirical data drawn from a survey of food webs with size-structure data. My comments follow.

**** Mathematical Model.** — Overall, there is not a sufficient description of the mathematical model in either the main paper or the supplementary information. I understand there are limits to what can be included in the former, but I think at least the defining equations, perhaps along a conceptual figure showing the schematics of the model, would be helpful to orient readers. In the latter, the authors several times refer readers to previous work, but I think there needs to be more. The analysis here should come closer to standing alone.

Our presentation of the SSSM in SI (S1 and S2) includes all defining equations. We double-checked that references to the underlying previous publication do not conceal information important for understanding our theory. However, we agree with the reviewer that in the original submission the flow of information through the model was difficult to understand. Following the reviewer's suggestions, we added a conceptual figure (Fig. S1), which shows how function-valued variables and parameters, i.e. those depending on body size, are linked in the model. This figure also serves as an illustration of the causal chains that work at fundamental level in the model.

— Perhaps in part because there isn't enough information in the SI, I was confused from the beginning as to what defines the effective growth rate for size class m^* . This seems to be clearly defined in Eq S1. But then another term is added in Eq S3, seemingly for pragmatic reasons, but I could also reinterpret S3 as simply redefining the growth rate $\Lambda(m^*)$, since it is linear in $B(m^*)$. How is this additional term to be interpreted ecologically/mechanistically? I don't quite understand whether/how it is in the model used to calculate $\Lambda(m^*)$ in the first place, and if it isn't, how we justify modifying one of the central equations in this way.

The role of the additional term is to suppress a model artefact that arises from model simplifications (a technique called “coarse graining” in the literature) that amount to the assumption that the strength of feeding interactions depends only on the body sizes of consumers and resources. When this dependence is continuous in both resource and consumer body sizes, this assumption implies

that species of very similar size have very similar consumers and resources and therefore compete strongly with each other. This strong competition among species of similar sizes means that the biomass in a narrow species size class can arbitrarily grow (or decline) while the biomass of a narrow neighbouring species size class declines (or grows), without otherwise affecting community dynamics, as long as the sum of biomass in the two species size classes remains constant. Minute irregular deviations from this neutral competition will lead to irregular increases and decreases of biomasses of species of similar size, resulting in a ragged species size spectrum that is smooth only when averaged over sufficiently wide intervals on the log-species-size axis. In reality food webs are more complex because traits other than body mass matter as well. Two species of similar size can therefore have very different sets of consumers and resources, and then do not directly compete with each other. Frequent competitive exclusion of species of similar size would only occur for (hypothetical) communities that are excessively oversaturated with species. Competitive exclusion would then lead to extinction of species until a (near) natural species richness is reached. Natural communities are in such a state: whenever competitive exclusion would happen, it has (mostly) happened already. In natural communities competition between species of similar size is therefore smaller than expected from the simplifying assumption that body-size alone controls feeding interactions. The correction term in Eq. (S3) puts this right by damping modulations of small wavelength along the species size spectrum.

Analytically deriving the form of this correction from first principles would require a better understanding of food-web structure and dynamic than we currently have. The correction term is therefore not included in the expression for $\Lambda(m^*)$ (and including it by force would not lead to pretty formulae). In Ref. (1) of SI [also cited as “CAT ”], heuristic considerations are presented for developing the specific form used here and choosing the two parameters entering it, which are ρ and σ_r in the parametrization chosen here. The value of ρ is constrained by a dimensional analysis alone. It has the same dimensions as the coefficients of the rates h and metabolic losses r . Combining these, one obtains a corresponding coefficient for the maximum rate of food assimilation minus losses, $\alpha h - r$. In our parameterization, this evaluates to $23 \text{ g}^{1-n}\text{yr}^{-1}$. From the dimensional analysis, one expects ρ to be of similar magnitude. Our choice $\rho = 10 \text{ g}^{1-n}\text{yr}^{-1}$ appears reasonable on this basis. The heuristic argument of CAT to constrain the width of the size range of damping σ_r suggests it should be of the magnitude of the typical distance on the $(\ln m^*)$ -axis between the main predators or prey of a species. Assuming that a typical consumer has approximately three prey species making a major contribution the diet (and fewer “main” predators), and noting that we set the width of the predator-prey mass ratio window to $\sigma_s = 1.5$, our choice $\sigma_r = 0.5 = \sigma_s / 3$ is consistent with these considerations.

We have integrated this additional discussion into the SI text, splitting it up between model definition (S1, lines 71-92) and parameterisation (S2, lines 142-153).

— The scheme for computing responses to press perturbations in Supplementary Sections S6 and S7 should be referred to more explicitly in the main text, as this is really the critical part of the analysis. I would suggest demonstrating the approach on a cut-down simplified version of the authors model, perhaps even a case (however unrealistic) where the responses to press perturbations can be computed analytically. This will give readers a better intuition for the overall approach, and for the approximations used e.g. in considering only the first few terms here.

We thank the reviewer for this suggestion. In writing the main text, we made efforts to highlight the existence of a mathematical theory underlying our analysis while avoiding a degree of exposure to this theory that would reduce attractiveness of the article to the general readership of Nature Communications. From the reviewer comment, we understood that existence of this theory, which is summarized in SI, was still not sufficiently clear in the main text. The key sentence in the original submission was

From previous work²⁷, it is known that pelagic size spectra exhibit a superposition of three mathematically distinct linear responses to pressure on a single body size class (Fig. 4, S6):

...

We have reworded this as (lines 152-154)

From a previous mathematical analysis²⁷ it is known that pelagic size spectra exhibit a superposition of three ~~mathematically~~ distinct linear responses to pressure on a single body size class (Fig. 4, Methods, S6): ...

Following the reviewer's suggestion, we have developed an analytically tractable example on which we demonstrate the method and the validity of the truncation of terms. This is now presented in Supplementary Information S6.4 and referenced in *Methods – Mathematical analysis* (lines 367-368).

** Conceptual advances.

— Many places in the text compare this approach/results to more general ideas of pattern formation across biology and elsewhere. Sure—there are analogs. But in my reading this is (a) too prevalent in the text. These examples do not add much and the current analysis basically stands alone. (b) not dealt with carefully enough where it could be useful. For example, the standard way (that at least I am familiar with) to think about Turing instabilities is as a result to small perturbations which are then allowed to relax. These take an unstable equilibrium solution towards (or not) a patterned, periodic solution, and the analysis proceeds by considering the eigenvalues of various modes, the sign of their real parts, etc. This is distinct from the press perturbations considered here, where one or more species abundances is essentially permanently reduced (or increased). This more likely corresponds to a case where a parameter in the model (like harvesting rate, as mentioned by the authors) is perturbed, as opposed to where abundances are instantaneously perturbed as in the case above. But these distinctions are glossed over to my mind, making the comparison with other pattern-forming systems confusing at best.

We agree with the reviewer that the method of analysis that we chose is different from the standard method used in the study of pattern formation. However, we disagree that the phenomenon itself is so different that this would invalidate our applications of the existing understanding of pattern formation to our problem. Invoking this existing understanding is useful for understanding the ecological problem. In particular, the two insights that (1) for amplifying cascades pattern amplitude is eventually controlled by inherent non-linear regulation (rather than pressures) and that (2) far from the onset of instability systems structure and dynamics tend to be controlled more by the phase of the pattern than its amplitude are important for the interpretation of field observations.

In *Method - Mathematical analysis* (lines 356-367), we are briefly comparing our approach to the analysis of the pattern-forming instability in size spectra with the standard method (outlined by the reviewer) and explain why the standard method is not applicable here (the base state is homogeneous along the log-size axis, but the time scales of dynamics depend on body size, so dynamics are not homogeneous and standard approach fails). We are also contrasting the standard criterion for identifying pattern formation (“When modulations grow through time, patterns form.”) with our criterion (“Patterns form when the equilibrium response increases with the distance from the perturbation along the size axis.”) and state that “In cases where both methods are applicable, they give equivalent results.” (emphasis added). We did not, however, provide evidence for this last statement, because we assumed it was intuitively clear. We are grateful for the reviewer's comment, which shows that more explanation of this point is required.

We therefore now refer in the discussion of the mathematical analysis in Method (lines 367-368) to the new Sec. S6.4 of SI, where we demonstrate that, indeed, the two criteria are equivalent for the

simple example discussed there, where both methods are applicable. There we also note that, while the method of analysis we use is suitable for describing the transition from convective stability to convective instability (corresponding to the transition from attenuating or amplifying responses to press perturbations), it breaks down when systems become absolutely unstable (for a brief explanation of these concepts, see <https://www.quora.com/What-is-the-difference-between-convective-and-absolute-instability>). The reason is that our method is based on the construction of stable steady-state model solutions, something that does not generally exist for absolutely unstable linear systems subjected to press perturbations.

These differences in analytic methods notwithstanding, the phenomenon we describe is a genuine case of pattern formation in the classical sense, not just a mere analogy. Michael Cross applied already in 1988 the classical mathematical apparatus of pattern formation to study stationary linear and non-linear patterns in convectively, but not absolutely, unstable systems [<https://doi.org/10.1103/PhysRevA.38.3593>].

— I would like to see a clearer set of links to other, existing work on top-down and bottom-up pattern formation in this context. The paper cites other work adequately, but I was left a bit confused about what would have to be omitted from this model to end up with (for example) only the possibility of top down cascades. In what limiting cases would the authors say their model reduces to other approaches? I also note a very recent paper which qualitatively reaches some similar conclusions (, *Ecology Letters*, 2018)

If there is no food to eat, certainly higher level consumers cannot survive. Correspondingly, if basal production declines abundances at higher trophic levels will tend to decline. This is a general ecological phenomenon, which implies that some level of bottom-up effects is always present. They might be attenuating in some models, but they are always present to some degree. We therefore cannot provide conditions for there being only top-down cascades.

We were unable to see the “similar conclusions” in Barbier and Loreau (2019, doi: 10.1111/ele.13196) in the relation to the questions we ask here. Barbier and Loreau (2019) are innovative with respect to different questions. Their “synthetic model developed in Box 1” does not seem to be much different from the model studied in detail by Heath *et al.* (2014, doi: 10.1111/ele.12200), which we cite and discuss as appropriate. Unfortunately, Barbier and Loreau (2018) do not refer to the work of Heath *et al.* The particular condition Barbier and Loreau (2019) derive for amplifying food top-down cascades (their Eq. 20) might be due to the short food chain they consider in the underlying calculation. For long food chains amplifying top-down cascades do not typically arise in models with Type I functional responses; some consumer satiation is required, see Section 21.3.3 of *Food Webs and Biodiversity* (Rossberg, A. G., Wiley, 2013). The analyses of Barbier and Loreau (2019) for the limit of infinite food chains overlap to a large part with Chapter 21 of this book. The difference is that they take the potential effects of inherent self-limitation into account, while Rossberg (2013) includes the effects of saturating functional responses. Barbier and Loreau (2019) do not appear to be aware of this work, either, and it appears that some of their conclusions would need to be modified if saturating functional responses were considered. Because the effects we describe in our ms depend essentially on saturating functional responses, it is highly unlikely that Barbier and Loreau (2019) arrived at conclusions similar to ours. We are now citing Barbier and Loreau (2019) in relation to food chains (line 5). As explained, the major innovation of the present manuscript is specifically that it overcomes the limitations inherent to the well-studied food-chain paradigm.

** Comparison with data— I was not totally sure how to evaluate how well the model performs against these data sets, but I do respect the effort the authors have gone to to assemble these data sets carefully and then perform the comparisons. Where I was stuck was this: the authors make quite a big deal of the fact that their approach can predict much more than highly simplified models, which might (say) predict domes, but nothing more precise than that. But at the same time, the authors’ own comparison with the data was only semi-quantitative, and I wasn’t

sure what would count as a poor vs a good fit. Basically—readers should understand how well the model did, beyond just eyeballing the fits.

In explaining our methodology, we refer to the concept of Pattern-Oriented Modelling, citing Grimm et al. (1996). As Grimm et al. explain, Pattern-Oriented Modelling aims to reproduce observed patterns in data (not the quantitative data itself) and the “scale” or magnitude of effects. The objective is to understand mechanisms, not to make quantitative predictions. This answers the reviewer’s question about what constitutes a poor vs a good fit: a good model fit is one that reproduces the major patterns in the data and gets the “scale” or magnitude of effects right. A poor model or model fit is one that fails to reproduce major patterns in the data or gets the “scale” or magnitude of effects wrong. Pattern-Oriented Modelling is less ambitious than quantitative predictive modelling. Good or poor fits can therefore be distinguished by eyeballing a graphical comparison of the characteristics of empirical data and model outputs. We rewrote text where we discuss the comparison of model outputs with data in Fig. 3 to make this clearer (lines 133-136).

Researchers generally quantify goodness of model fits for two reasons: (1) to determine the ability of model to make quantitative predictions and (2) to quantitatively compare the ability of one model to fit the data with that of another, ultimately for the purpose of model selection. But we do not aim at quantitative predictions, and there is no competing model that reproduces the set of patterns we consider even in the weak sense of Pattern-Oriented Modelling. The model selection procedure implicit in our work therefore does not require a quantitative measure of goodness of fit. Comparison of the patterns that models are capable of reproducing is sufficient. This comparison we provide in the paragraph starting “The analytic calculations also reveal why models simpler than the SSSM would struggle to convincingly explain dome formation.” in *Discussion - A mechanistic explanation* (lines 249-267).

One might wonder whether it would nevertheless be useful to quantify goodness of fit of the SSSM, e.g. for the purpose of comparison with some other more advanced model in the future. However, we would caution against such a comparison. Model selection methods based on quantitative goodness of fit criteria draw on the assumption of there being a well-defined statistical ensemble (or a family of ensembles spanned by a few system parameters) that the models aim to approximate. The size-spectrum data currently available, however, are too inhomogeneous in both biological conditions and methods to be representative, in a statistical sense, of a well-defined ensemble. Quantitative model selection methods would overstretch the interpretation of this data. It is therefore prudent not to compute quantitative measures of goodness of fit for these data.

Modelling with the objective of merely reproducing observed patterns, rather than quantitative fits, is nothing unusual in ecology. The reviewer’s impression that the SSSM might even be capable of quantitatively fitting the data is encouraging, but the question whether this is so is not central for our argument.

This is compounded by the fact that the model actually fitted to the data (Eq 1 in the main text) is if I understood correctly itself a phenomenological fit from simulated data. I.e. not actually an analytical outcome of the model, but instead an approximation that works well (in some? all?) parameter regimes. So if this phenomenological fit fails, does that mean that the underlying, full model fails? Or could Eq 1 sometimes fail to fit both empirical and simulated data in the same ways. This could be made clearer.

We are using the fit of the phenomenological model give by Eq. (1) to data simply as a means to characterize and compare empirical data and data generated by the SSSM. In the sentence preceding Eq. (1) (lines 109-110), we now write “To characterise the main features of the spectra ...” rather than “To quantify the main features of the spectra ...” to make this clearer.

Formally, the phenomenological fit simply acts as a black-box machinery that takes empirical and modelled size-spectrum data as inputs and generates four numbers that depend on this data as output. By the fact that the four numbers depend on the data and nothing else, they are characterizations of the data. The phenomenological fit therefore cannot “fail” to fit the data. The particular choices of the model in Eq. (1) and of the fitting procedure result in four quantitative characterizations that have an intuitive interpretation and that exhibit patterns in the dependence of size spectra on nutrient availability that are, as we argue, difficult to reproduce using simpler models. These properties justify our particular choice of Eq. (1). Most other conceivable ways of computing four numbers from size-spectrum data sets would not have these properties (and would thus “fail” in this sense). For an in-depth discussion of the underlying general idea and argument, please see Chapter 2 of *Food Webs and Biodiversity* (Rossberg, A. G., Wiley, 2013) or <http://axel.rossberg.net/paper/Rossberg2007d.pdf>.

Reviewer #2 (Remarks to the Author):

Review of manuscript: “Dome patterns in pelagic size spectra reveal strong trophic cascades”
Overall comments Using empirical data, simulations and the existing framework of Species Size Spectrum theory the authors offer a novel interpretation on observed domes in biomass formed across a wide size spectrum. Existing literature suggests that domes form as a result of a bottom up effect whereby each subsequent dome in biomass of a specific size spectrum arises as a result of the dome on its left side. The authors reconstruct the framework suggesting the opposite: that it is the higher predators that essentially drive the dome formation through top down effects, and this effect is pronounced under eutrophic conditions. Within this framework, apparent contradictions within food chain as well as size spectrum theories seem to be resolved.

I can see this approach having its merits for ecosystem modelling which typically tends to over simplify a system by focusing only on its emergent properties of eg energy flow and nutrient cycling by averaging out important interactions between species populations and traits. This essentially “black box” modelling approach often compromises flexibility (i.e. maintaining variability in space and time as conditions change). This manuscript might offer a way out: Knowledge on the size domes represented in a specific ecosystem and the reasons leading to the emergence of these domes could in the future replace the rigid “black box approach” in representing biota, still without overcomplicating models by including foodweb interactions and their strengths. I believe the idea presented here is important and is offering a refreshing new perspective for ecological theory underlying foodwebs. The figures offer useful conceptualisation for understanding the patterns and mechanisms. Having said that, I believe that certain points need improvement and specifically:

We would like to thank the reviewer for highlighting the key merits and points of this study and for the valuable feedback.

1. The introduction must be more focused avoiding redundant information

We edited the introduction according to the reviewer’s suggestions under *Specific comments* below. We also substantially shortened the discussion of lumpy coexistence theory (lines 62-64).

2. A description and ecological implication of the size spectrum characteristics should be provided

In the paragraph introducing our method for characterizing size spectra, we added direct ecological interpretations of the four size spectrum characteristics S , B_0 , A , and D (lines 112-124).

3. The mechanisms must be explained in a more detailed manner

We have now emphasized in the main text that the detailed explanation of the mechanisms is in Supplementary Information S7 (lines 163-164, line 171). The detailed explanation is too long to include it in the main text.

4. A more careful discussion on the implications of the relevance of this framework for coastal ecosystems should be attempted.

We added a note in the text stating that the range of chlorophyll-*a* concentrations where we predict dome formation in coastal marine waters is below the levels at which harmful algal blooms tend to occur (line 240).

I also found that the inclusion of lumpy coexistence theory in the introduction was unnecessary for reasons I explain below.

It is important for our argument that we induce the concept of self-organised pattern formation in trait space, because an understanding of dome formation as an instance of such pattern formation implies that the observation in size spectrum models of modulations along the size axis alone is insufficient justification for the conclusion that the mechanism generating domes has been identified (many different mechanisms can lead to such modulations). Since we therefore unavoidably discuss this topic, we would consider it inappropriate to disregard the literature on lumpy coexistence, which might be better known to many readers. However, we agree that our discussion of lumpy co-existence was more extensive than necessary. We have now condensed it into a brief sentence referencing some of the relevant literature (lines 62-64).

Specific comments Summary

Line 18: replace by: “Compiling size spectrum data of high-quality...”

This has been corrected (line 18).

Lines 22-25: this sentence is very confusing as it does not follow from the description of the study remit as given above but rather seems to convey an altogether different message that is confusing for the reader.

To make the sentence less confusing, we have now removed the physics jargon, leaving essentially only the information that we do make use of interdisciplinary arguments and what the ecological context is in which they are applied (line 22-24).

Line 71: not all of these models are abstract though. Some have an empirical basis by relying on experimental and field data for parameterisation. Thus these sentences must be re-thought.

We have removed the sentence, because it was slightly off topic.

Introduction

The first two paragraphs need to be more focused and less abstract, connecting better with the summary and aims. Avoid being unnecessarily wordy and connect the two paragraphs into one with a stronger opening line and a punchy last sentence indicating the challenge.

We have followed this advice (lines 25-40).

Lines 26-28: the opening statement is too vague...what is implied by “concepts” in this context? Theories perhaps, frameworks?

There probably were simply too many ideas covered in the first sentence. The dictionary definition of “concept” is an abstract idea of how things are, and this is what we mean. We have simplified this sentence and made it clearer (lines 25-26).

Lines 28-29: My understanding was that a food chain consists of only one organism per trophic level as opposed to food webs which represent groups of organisms per trophic level, or else an ensemble of many food chains...

There is this notion of “food chain” used in the literature, but also the notion of “the” food chain, consisting e.g. of producers, herbivores, carnivores (eating herbivores) and carnivore-eating super-carnivores. Of course this is an abstraction, but it is being used, please see the references we cite.

We refrained from discussing this distinction in our manuscript, but we have made sure there is no comma after “food chains” in “food chains formed by groups of organisms assigned to discrete trophic levels”, thus grammatically permitting existence of other kinds of “food chains” (lines 26-28).

Line 30: “where pressures on top predators”, do you actually mean, “pressures of top predators”?

We mean “pressures on top predators” (now line 29). This could be fishing, diseases, extra feeding, etc. All that is required to set the mathematical machinery of top-down cascades into motion in food-chain models is some kind of press perturbation at the highest trophic level.

Line 32: Their strengths vary, instead of varies

This has been fixed (lines 30-31).

Line 46: This paragraph is badly connected to the paragraphs above: (1) an alternative perspective to what? To the food chain structure mentioned in the first paragraph? Need to stress the connection. (2) Also please explain why there needs to be an alternative in the first place...eg in order to elucidate, avoid the aforementioned discrepancies..., (3) in which way does this theory help overcome the problems raised above? Be more explicit

(1) We have reworded this sentence and now write (lines 41-43) “An alternative way of looking at the structure of an ecological community...”. (2), (3) Ecologists have historically been measuring size spectra (since at least 1972) and found this interesting. There was no specific motivating need. We decided not to recall the history of the field in the ms.

Line 56: “For the left-most dome, other explanations are required”...such as? Nutrient control? Offer an example

We now say “another explanation would be required” (lines 51), to emphasize the aspect of uncertainty. It’s not our theory what we describe here. We just note that the explanation of domes that the bottom-up theory offers does not work for the left-most dome (corresponding to the lowest trophic level). This could be a major weakness of the bottom-up theory of dome formation, but then again there may be a very simple explanation. We find it difficult to criticise or constructively complete a theory that has never been fully formulated in a process-based dynamic model and of which some variants are inherently inconsistent even by the authors’ own admission (Thiebaut & Dickie, 1993, p. 1315).

Line 66-73: I’m really not sure how this paragraph is relevant to the rest of the paper. This paragraph discusses theory on lumpy co-existence which has been supported by mathematical theory (eg Turing’s work), abstract consumer resource models, and resource competition models parameterised on experimentally established species traits. Therefore, although the authors are correct in saying that there are no substantial empirical evidence to prove this theory (although limited evidence is presented within Scheffer & Van Ness PNAS 2006), their claim that all models used so far are abstract is incorrect. But more importantly, this theory is referring to species that are competing for similar resources (i.e. species of the same assemblage) and troughs in trait space arise a result of competitive exclusion. Thus, I’m not sure how this conceptual framework can be generalised to describe different trophic levels along a trait space. Moreover, the lumpy coexistence framework suggests that the y axis represents number of species within a certain size class that are favoured as a result of evolutionary processes or long term ecological processes acting on the assemblage. This is conceptually different to the size spectrum theory which suggests that specific size ranges develop more biomass, which, to my understanding, does not preclude the existence of rare species in the troughs.

We refer to lumpy co-existence (lines 62-64) because it is another and perhaps better known case of pattern formation in trait space. Following the reviewers’ comments, we have now condensed the

discussion of lumpy co-existence into a brief generic sentence referencing some of the relevant literature.

Both biomass and number of species have been used for the y-axis in studies of lumpy co-existence (see Scheffer & Van Ness PNAS 2006, Figs. 2 and 3).

Line 93: instead of “mechanical” do you mean “mechanistic”?

Yes. We now write “basic mechanism” (line 73).

MODEL STRUCTURE AND BASIC BEHAVIOUR Line 94: it is not clear what is meant by “abstract but complete description of life histories“

Thank you. This sentence has been removed.

100-108: this naturally leads to the question of what would happen if a species from a middle dome was removed...

We agree that this is an interesting question, but the answer does not immediately contribute to clarifying whether domes form through a top-down or bottom-up process in the SSSM. Due to space limitation, we cannot address the question in the manuscript. However, we provide together with the manuscript our simulation code, which permits interested readers to conduct such experiments within a few minutes once the code has been compiled.

Figure 2: “dome size is controlled through internal mechanisms rather than cascading pressures.” Not clear what is meant by internal mechanisms here

We now write “inherent non-linear regulation” throughout the ms and added (Fig. 2, line 184) references to Cross (1988), where such inherent regulation limiting pattern amplitude is demonstrated for a convectively unstable but absolutely stable pattern-forming system (lower panels in Cross’ Figs. 3, 4).

It is clear that when the amplitude of domes becomes larger and larger the linear approximation underlying Fig. 4 of our manuscript will eventually break down. Non-linear effects need to be taken into account. For most pattern-forming systems these effects are to lowest non-linear order such that they prevent further increase of the pattern amplitude (i.e. an increase in time or, in our case, in the direction of propagation of perturbation responses). The equilibrium amplitude is then given by the value where linear growth (in time or in the direction of propagation) and non-linear suppression balance each other. These are general principles of non-linear pattern formation, independent of the detailed mechanisms that generate patterns. They can be understood without knowledge of the detailed linear or non-linear mechanisms at work. This is why we know that inherent non-linear regulation will eventually control dome size in our model, without knowing the detailed non-linear mechanisms at work.

The structure of this argument is similar to that used by ecologists in arguing that an exponentially growing population will eventually be limited by some density-dependent effect. Ecologists therefore extend the equation for exponential growth $dB/dt = r B$ by a term describing density dependent regulation. To lowest non-linear order in B , this term takes the form $-c B^2$, thus leading to the logistic equation $dB/dt = r B - c B^2$. Ecologists have studied the solutions of this equation and variants of it and fitted it to data without a need to know in detail what the ecological mechanisms controlling the value of c are (which might be some combination of resource limitation, interference competition, control by diseases, etc.).

Paragraph starting line 121: although the size spectrum characteristics are mentioned here, a description of what they actually represent in a biological community should be provided. This would also help readers better understand the mechanisms affecting the patterns across the size spectrum when eutrophication level changes.

We have now included ecological interpretations of these characteristics in the text (lines 112-124).

MECHANISMS

I believe that figure 3 should become more self-explanatory by including a short description of each characteristic of size spectra.

We agree with the reviewer and have added a sentence explaining the four characteristics in the figure caption.

159-161: it is unclear how the increase in the slope of the size spectrum would lead to an increase of abundance of consumers relative to resources. This needs more explanation to make it explicit

We added “Because consumers tend to be larger than their resources, ...” (lines 166-167).

167: Both enhance, instead of enhances

Fixed (line 174).

156-168: I find this paragraph rather problematic in describing the mechanism behind the effect of increased eutrophication on the troughs and domes. More care should be given to explain exactly what changes in community structure lead to the troughs.

We strongly sympathise with the sense of dissatisfaction by the reviewer with this explanation. Indeed, the explanation is incomplete and full of gaps. Our complete (“exact”) explanation is given in SI Section S7 (building on the existing theory recalled in Section S6) and stretches over 8 pages. If there was a way to provide the complete explanation in less space then we would have done so. Now, if the complete explanation requires at least 8 pages, then every summary of the explanation that is shorter must be incomplete and contain gaps. Attentive readers such as the referee will always notice these gaps, which is why we refer to the full explanation in SI. We have added in this paragraph words highlighting that the details of the explanation can be found in S7 (lines 164, 171).

Discussion

line 216: rephrase

We have corrected the grammar and simplified the wording (line 226).

lines 222-223: These considerations

We reformulated this sentence (lines 232-234).

line 220: “nonlinear mechanisms inherent to the system”, I’m very intrigued by this phrase, please elaborate

We reworded this to “inherent non-linear regulation” (line 230) for consistency with Fig. 2. We have also included the references to Cross (1988) and Newell et al. (1993). Please see our response to the corresponding comment regarding wording in the caption of Fig. 2. In brief, we can expect this non-linear regulation to exist on general grounds without knowing the detailed mechanics of it.

Conclusions:

Lines 270-282: Most of this paragraph is philosophical and redundant. I would omit/merge with next one.

We fully agree with the reviewer that this paragraph (now lines 280-292) presents philosophical considerations. We might be in disagreement with the reviewer about the usefulness of philosophical thought. We believe that it is important to prepare the subsequent discussion by recalling the general principles of the relation between high-level and low-level accounts of emergent phenomena, which leads to apparent paradoxes such as that high- and low-level accounts can be very different and yet describe the exact same thing, and that high-level accounts can be useful despite being in some sense “wrong”. We do not expect the general reader to be sufficiently familiar with these considerations, and therefore recall them in this paragraph.

283-293: since we are limiting this to the prediction of community structure in eutrophic coastal ecosystems, we should also consider that most often eutrophication lead to HABs which selectively decrease the abundance of organisms up to the top predator level.

It appears that this comment refers to the 5th paragraph in *Discussion – A mechanistic explanation* predicting dome-formation in coastal marine water. We have added text (line 240) noting that the chlorophyll-*a* concentrations in the range of interest ($0.2\text{-}1.6\mu\text{gL}^{-1}$) are lower than the levels at which harmful algal blooms tend to occur. That is, what we are talking about here are coastal waters that are nutrient rich but not *that* rich.

Reviewer #3 (Remarks to the Author):

Dome patterns in pelagic size spectra reveal strong trophic cascades
Pavel Kratina

Axel G. Rossberg, Ursula Gaedke and

In aquatic ecosystems, the living biomass as a function of organisms' size is distributed according to quasi-linear log-log size-spectra “from bacteria to whales”. A common and well-known feature of these size-spectra is their linearity. However, in some specific ecosystems, and this has been particularly observed in lakes, size-spectra can exhibit distinct stationary domes (e.g. Yurista et al., 2014). In this paper, Rossberg and co-authors attempt to explain these domes using a size-structured community model where an infinite number of species distinguished by their size at first reproduction experience size-based trophic interactions. The mathematical model developed and used is indeed displaying periodic bumps of increasing amplitude when nutrient enrichment increases (Fig. 1). It is shown that these patterns emerge from bottom-up effects when the abundance of top predators increases (Fig. 2) due to nutrient enrichment (Fig. 3). While the model proposed is undoubtedly very sophisticated from a mathematical and numerical point of view and thus very interesting for applied mathematicians and theoretical ecologists, I have not been convinced by the ecological conclusions inferred from its analysis and simulations and the discussion of these results. The major issues I had when reading this paper are listed below:

- The domes observed empirically have been shown to match well-identified taxa and functional communities (e.g. viruses, heterotrophic bacteria, unicellular phototrophs, unicellular mixotrophs and heterotrophs, planktonic crustaceans, fish). Each taxa/functional community occupies well-defined and successive size ranges over 3-4 orders of magnitude (e.g. Boudreau and Dickie, 1992; Sprules and Goyke, 1994; Thiebaut and Dickie, 1993;) due to different limiting processes occurring in these size ranges (Andersen et al., 2016). This is the standard paradigm at the moment and Rossberg et al. don't discuss it and don't explain why it should be abandoned and replaced by a model that is not consistent with the empirical fact that the size ranges of dome-specific taxa don't overlap (in the model, birth size is the same for every species so that the species present in a given dome overlap with all the species present in smaller domes).

We agree with the reviewer that domes are often labelled taxonomically in the literature, e.g. “phytoplankton”, “zooplankton”, “(planktivorous) fish”. And, for instance in Boudreau and Dickie (1992), there is scope for reading this attribution of domes to taxa as a causal explanation. But we do not find that the literature cited by the reviewer is always clear about this question. Yurista *et al.* (2014), for example, attempt to predict the height and position of the “Predatory Fish” dome NOT from the size of typical predatory fish, but by extrapolating the periodic structure of the dome pattern from the “Prey Fish” dome to the “Predatory Fish” dome. Taxonomy or different limiting processes for “Prey Fish” and “Predatory Fish” do not play a role in this calculation. We therefore

do not agree that the interpretation favoured by the reviewer is necessarily the “standard paradigm”. From our reading of the literature, the standard paradigm is the explanation of domes as resulting from a bottom-up cascade (see literature cited in 2rd paragraph of introduction).

On the other hand, Andersen et al. (2016) do indeed note a number of differences in the ecology of groups of organisms occupying different size ranges. But their review relates mostly to marine ecosystems, where dome patterns are weak or absent. Consequently, Andersen et al. (2016) do not attempt to explain dome patterns by these size-dependent differences in ecology. The interpretation favoured by the reviewer raises the question why dome patterns are weak or absent in marine system.

Nevertheless, the reviewer’s view that of dome patterns are controlled by body-size specific limitations on ecology or biology of organisms merits consideration. In fact, this view is not necessarily inconsistent with our finding. As is known from the theory of non-linear pattern formation, the phase of spatial modulation patterns (here the position of domes along log-size axis) can be affected by weak inhomogeneity of the system (in our case, small deviations from allometric scaling laws for biological or ecological traits of organisms). If there are size ranges with particularly favourable biological or ecological traits, domes are expected to exhibit a tendency to position themselves in these ranges, as long as this is compatible with the main mechanism generating domes. Conversely, if species in some size ranges tend to have unfavourable traits, the positions of domes can be expected to have a tendency to avoid these. In a brief appendix below, we show the results of preliminary model simulations [redacted]. We decided not to include this material in the manuscript; as it stands, it might raise more questions than answer, and space constraints do not permit going into details. Study of this phenomenon will be the next step in developing our understanding of domes patterns. However, we

have added a sentence at the end of *Discussion – mechanisms* (lines 190-192) stating: “This sensitivity of dome positions to pressures might also lead a tendency for domes to form in biologically favoured size ranges, for which there is some empirical evidence [Sprules et al. 1983].”

With reference to other points in the reviewer comment, we may note that the size ranges of dome-specific taxa usually *do* overlap. If they would not overlap there would *always* be gaps (without any individuals of that size) in size spectra between domes. What is observed is that gaps are the exception rather than the rule.

Finally, it is not correct that our model assumes that birth size is the same for every species. In the SSSM body mass at birth for each species is x_0 times its body mass at first maturation, with $x_0=0.02$. However, Rossberg (2013) has shown that the effect of changing x_0 on the dynamics of the species size spectrum is small (the effect on the resulting community size spectrum is larger).

• The domes observed empirically are stationary: they don’t propagate through the size-spectrum. On the contrary, and unless they explicitly include stabilizing density-dependent processes, size-spectrum models are known to be prone to unstable behavior leading to the formation of traveling waves (i.e. non-stationary domes) that are propagating through the spectrum (Datta, 2010; Hartvig et al., 2011; Plank, 2012; Plank and Law, 2012; Maury and Poggiale, 2013). Despite incorporating an artificial stabilizing term (eqs. S3), the same behaviour is mentioned (but never clearly described) for the model presented when $x>10$ so that only snapshots are shown in the paper (e.g. Fig. 1). This casts doubts on the homology that Rossberg et al claim between the model behaviour (i.e. unstable traveling waves) and the ecological processes responsible for the presence of stationary domes.

We agree with the reviewer that travelling waves often found in size-spectrum models are inconsistent with observations, at least when these waves are identified with domes. But the SSSM does not generate travelling waves. To linear order in dome amplitude, the size-spectrum modulations in the SSSM are stationary, as seen empirically for domes. What “travels” is only the

edge of the size range over which modulations of the size spectrum extend. The dynamics previously setting in for $x > 10$ have become much less pronounced after we corrected the bug in our simulation code. In parameter range where it persists (very high x) it remains of a kind that is different from travelling waves. Instead, domes initially wobble up and down and left and right, and in some cases the wobbling can become so strong that individual domes may disappear and reappear. We are unsure about the question to what extent this dynamic phenomenon has a counterpart in nature, and believe that the currently available data is insufficient to settle this question. Certainly we do not expect a 1:1 correspondence. Reasons include the real-world complications that the reviewer mentioned above and that natural size-spectrum dynamics are complicated by seasonal variation, which is not currently included in our model. This is why we do not describe the details of the dynamics for high x in the manuscript. The variation in the overall structure of size spectra resulting from these dynamics, captured by the areas in pink in Fig. 3, is rather small.

- The presentation of the model in the paper and the supplementary material is confusing and insufficient to understand exactly how this model is constructed, simulated and used.

We agree with the reviewer that the SSSM is more complicated than previous models and that our simulations invoke some advance (e.g. pseudospectral) methods, known mostly outside the ecological community. However, we also believe that the difficulty of the problem, which previous work using simpler approaches has revealed, warrants the introduction of advanced methods.

As we wrote in response to Reviewer 1, we have now added a graph in SI (Fig. S1) that illustrates the flow of information through the model and should make the underlying ecology more transparent.

Overall I believe that the claim that this model explains the dome patterns observed has not been unambiguously and convincingly demonstrated in this paper. Furthermore the model presentation is not really understandable as it stands in the paper and sup. mat. and the discussion of the results in an ecological perspective is insufficient (why should we abandon the standard explanation for the domes?). Lastly, the focus of the paper is maybe too narrow for a generic journal as Nature Communication and I would therefore recommend re-submission in a more specialized journal after the concerns expressed here have been addressed.

We believe that it is especially the interdisciplinary nature of this work that makes it interesting for the broad readership of Nature Communications. We combine traditional ecological thinking, the generic understanding of non-linear patterns developed by physicists, advance numerical techniques, novel mathematical ideas, and a philosophical perspective to develop and support our conclusions. We hope that, as an interdisciplinary contribution, it invites readers from a diverse range of backgrounds to appreciate the beauty of the unity of science.

We have responded to the other concerns above.

Reviewer #3 references:

Andersen K. H., T. Berge, R. J. Gonçalves, M. Hartvig, J. Heuschele, S. Hylander, N. S. Jacobsen, C. Lindemann, E. A. Martens, A. B. Neuheimer, K. Olsson, A. Palacz, A. E. F. Prowe, J. Sainmont,

S. J. Traving, A. W. Visser, N. Wadhwa, and T. Kiørboe, 2016. Characteristic Sizes of Life in the Oceans, from Bacteria to Whales. *Annu. Rev. Mar. Sci.* 2016. 8:217–41.

Boudreau, P.R., Dickie, L.M., 1992. Biomass spectra of aquatic ecosystems in relation to fisheries yield. *Can. J. Fish. Aquat. Sci.* 49 (8), 1528–1538.

- Datta, S., Delius, G.W., Law, R., 2010. A jump-growth model for predator-prey dynamics: derivation and application to marine ecosystems. *Bull. Math. Biol.* 72 (6), 1361–1382.
- Hartvig, M., Andersen, K.H., Beyer, J.E., 2011. Food web framework for size-structured populations. *J. Theor. Biol.* 272 (1), 113–122.
- Maury, O., Poggiale, J.-C., 2013. From individuals to populations to communities: a dynamic energy budget model of marine ecosystem size-spectrum including life history diversity. *J. Theor. Biol.* 324 (1), 52–71.
- Plank, M., 2012. Effects of predator diet breadth on stability of size spectra. *ANZIAM J.* 53 (0).
- Plank, M.J., Law, R., 2012. Ecological drivers of stability and instability in marine ecosystems. *Theor. Ecol.* 5 (4), 465–480.
- Sprules, W.G., Goyke, A.P., 1994. Size-based structure and production in the pelagia of Lakes Ontario and Michigan. *Can. J. Fish. Aquat. Sci.* 51 (11), 2603–2611.
- Thiebaux, M.L., Dickie, L.M., 1992. Models of aquatic biomass size spectra and the common structure of their solutions. *J. Theor. Biol.* 159 (2), 147–161.
- Yurista P. M., D. L. Yule, M. Balge, J. D. VanAlstine, J. A. Thompson, A. E. Gamble, T. R. Hrabik, J. R. Kelly, J. D. Stockwell, and M. R. Vinson, 2014. A new look at the Lake Superior biomass size spectrum. *Can. J. Fish. Aquat. Sci.* 71: 1324–1333.

Appendix

REDACTED

Reviewers' Comments:

Reviewer #1:

Remarks to the Author:

I appreciate the authors' responses and edits. I am now following up on two of my earlier comments, and adding one additional request for clarification.

** "Correction term" for growth rate for a given size class.

I appreciate the authors' explanation—that there is some unknown "noise" in the interactions here between individuals with very similar masses. This noise arises from (unmodeled) differences between species/individuals of the same size, and there is then a kind of filtering arising from community assembly over time. If I understand correctly, the combination of both of these effects leads to slightly lower levels of competition between species of similar size (than would be found in the simplest version of the model).

My only additional comments related to this are

(i) This motivation did not come across to me in the earlier draft, and the new text in the rebuttal and SI is very helpful. I think that at least some of this informative explanation belongs in the main text too. Or at the very least a reference to the fact that the authors incorporated species differences and assembly in this way, and then a pointer to where more detailed discussion can be found in the SI. (One comment: I am not sure I would call this "coarse-graining", except in a fairly generic sense. This is maybe all the authors mean—in which case, fine).

(ii) There are interesting parallels with literature focusing on clusters arising from competition mediated by an arbitrary trait or traits, and the kernel determining competition strength between species with differing traits. The kind of correction here seems analogous to that discussed in (e.g. Fig 2 of) Barabas et al (2013) "Emergent neutrality or hidden niches?" *Oikos*. Is that a correct interpretation?

Similarly, the kind of "noise" (i.e. species differences) implicit in deriving this correction is analogous to the noise introduced in D'andrea et al (2018) "Translucent windows: How uncertainty in competitive interactions impacts detection of community pattern". *Ecology Letters*. Also correct?

Some/a couple of sentences relating to the literature on patterns, clustering and species differences in competition may be informative.

** Comparison with data and use of phenomenological model.

I appreciate the authors additional text and framing in terms of Pattern-Oriented Modeling. I do also agree that the patterns of the four parameters defined in Eqn 1 change qualitatively with $\log(TP)$ in the same way in both theory and data.

However, I am still finding the motivation for using this parametric form slightly difficult to follow. For example, would it not be cleaner just to evaluate directly total biomass predicted in the theoretical model, and total biomass in the empirical data, and show how these quantities vary with $\log(TP)$? Or to consider maybe some version of a Fourier transform and extracting the dominant frequency, rather than imposing the sinusoidal form?

There may be good reasons for using the parametrized phenomenological model, but readers may

need more help to understand these.

** One additional question which does not build on my previous review. In Fig 2a a simple demonstration is given of top down control in (some nutrient regimes of) the model. The top dome is depressed, and as a result there is an oscillating ("staircase") effect on other biomass levels in the community. In fact, for the H value chosen it seems as though the domes are lost altogether, but presumably for smaller H the domes would be present, and diminished in amplitude.

This all makes sense to me. However, one subtlety that may be helpful to clarify. Suppose that I thought (as per literature cited, L49) that "domes represent subsequent members of the aquatic food chain". Then I would probably look for the effects of top down control in alternating effects on the *heights of the domes themselves*. I.e. the classic staircase pattern that the authors show has upwards steps in the troughs, and downward at the domes. But if I think of the domes as members of a food chain, I am probably expecting the staircase effect to alternate across domes. In simple terms, I am wondering whether readers will expect the cascade arising from top down control to have frequency equal to the frequency of the existing biomass modulation (the authors' definition) or frequency equal to twice the frequency of the biomass modulation (as per my naive argument above).

This maybe is just another example of where the authors think (and are showing) that the interpretation above of domes as representing members of a food chain is incorrect, or incomplete. Some brief discussion might be informative.

Reviewer #2:

Remarks to the Author:

I believe the authors addressed the comments in a manner that benefited the manuscript and clarified previously confusing points.

Reviewer #3:

Remarks to the Author:

The revised paper by Rossberg and co-authors presents interesting results and stimulates curiosity. The key result, i.e. the fact that dome patterns in pelagic size spectra would emerge from top-down trophic cascades, is appealing (see fig. 1), even if a better discussion of alternative hypothesis such as outlined in my previous comments (discussion partly provided by the authors as a response to my comments but not included in the paper) would still be needed. The prediction that the dome structure increases with the input of nutrients into the system, and therefore mostly appears in eutrophic lakes, also seems to match observations (Fig. 3, S4.1).

However, the model that underpins the entire analysis is not presented in a way that most readers (including the aquatic / marine ecologists and ecosystem modellers that should be the primary target of the paper) could understand. This has not to do with the complexity of the mathematics used; it is a problem of clarity and completeness of the explanations provided as well as a lack of pedagogy. Indeed, almost no information about the model is given in the main paper and the supplementary material is far from being self-standing, despite the addition of fig. S1 (that unfortunately doesn't really help). It is a pity since the results presented would deserve a better justification and the model, which is claimed to be better than previous ones, would also certainly deserve to be presented clearly. Unfortunately this is not the case and at the moment the reader is asked to believe the results without

understanding them, and has no choice but accepting the unsupported claims that the SSSM model used in this study is better than previous models in the field and that the secret mechanisms it includes are sufficient to explain the patterns observed.

But what is different from previous models in the SSSM? What are the mechanisms included that are not included in previous models and how are they included? Is the model a trait-based model based on species-specific McKendrick–von Foerster equations like most other recent size-spectrum community models? Why not writing it explicitly? Why trait-based models such as the one in Hartvig et al. (2011) (which is said to be similar to the SSSM used here) don't generate such dome patterns? Is it because they are not integrated over a wide enough size range? Why SSSM don't generate traveling waves while Hartvig's model does? Why showing snapshots and not stationary solutions?

Even if the SSSM model is complicated, it has to be presented clearly and pedagogically, in simple terms, introducing the key equations and explaining their relationships without entering technical calculations. This is unfortunately not the case at the moment and the reader is only left with a very vague idea about what the SSSM actually is. Not only is this frustrating, it is also not acceptable in a paper whose central arguments rely on SSSM simulations.

Response to reviewer comments
Dome patterns in pelagic size spectra reveal
strong trophic cascades

(NCOMMS-18-30675B)

We thank the three anonymous reviewers for their careful reading and useful comments on our manuscript. In the following, we retained feedback that we received indented and in blue, and added our responses in black.

Reviewers' comments:

Reviewer #1 (Remarks to the Author):

I appreciate the authors' responses and edits. I am now following up on two of my earlier comments, and adding one additional request for clarification.

** "Correction term" for growth rate for a given size class.

I appreciate the authors' explanation—that there is some unknown "noise" in the interactions here between individuals with very similar masses. This noise arises from (unmodeled) differences between species/individuals of the same size, and there is then a kind of filtering arising from community assembly over time. If I understand correctly, the combination of both of these effects leads to slightly lower levels of competition between species of similar size (than would be found in the simplest version of the model).

My only additional comments related to this are

(i) This motivation did not come across to me in the earlier draft, and the new text in the rebuttal and SI is very helpful. Ellen?!: systematic differences between uni- or bidirectional trait axes (cf. Glossary) possible?!

I think that at least some of this informative explanation belongs in the main text too. Or at the very least a reference to the fact that the authors incorporated species differences and assembly in this way, and then a pointer to where more detailed discussion can be found in the SI. (One comment: I am not sure I would call this "coarse-graining", except in a fairly generic sense. This is maybe all the authors mean—in which case, fine).

We agree with the reviewer. In the new section "Model structure", we do not only mention that we included this correction term, but also motivate it in the context of size-spectrum theory. As the reviewer suggested, we also point the reader to SI for more detailed discussion.

Regarding "coarse-graining": yes, we mean this in a wide sense as used in the two references below:

<https://www.sciencedirect.com/science/article/abs/pii/S0375960117310587>

<https://royalsocietypublishing.org/doi/full/10.1098/rsta.2016.0338>

(ii) There are interesting parallels with literature focusing on clusters arising from competition mediated by an arbitrary trait or traits, and the kernel determining competition strength between species with differing traits. The kind of correction here seems analogous to that discussed in (e.g. Fig 2 of) Barabas et al (2013) "Emergent neutrality or hidden niches?" *Oikos*. Is that a correct interpretation?

This is an interesting observation by the reviewer. We agree that Barabas et al (2013) describe a similar correction with a similar motivation. We mention this now when we discuss the correction term in the main text. But we also note an important particularity of our approach, which is that it avoids unaccounted for biomass losses.

Similarly, the kind of “noise” (i.e. species differences) implicit in deriving this correction is analogous to the noise introduced in D’andrea et al (2018) "Translucent windows: How uncertainty in competitive interactions impacts detection of community pattern". Ecology Letters. Also correct?

We are not sure if this idea is correct. D’Andrea et al (2018) mention different kinds of noise (process, measurement), and in our case we need to add to these numerical noise and environmental variability/stochasticity due to variations in community structure. Because neither Barabas et al (2013) nor D’Andrea et al (2018) nor Rossberg (2013) derive these correction terms analytically, it is not clear how exactly all these different sources of variability come together to produce the observed effect. We do now mention D’Andrea et al (2018) in the manuscript, but decided not to develop these speculations. This should be topic of another, systematic study.

Some/a couple of sentences relating to the literature on patterns, clustering and species differences in competition may be informative.

It is there in the last paragraph of introduction, but perhaps not sufficiently obvious. We now explicitly mention “competition” in the relevant sentence, saying “It has long been speculated that such structures can arise not only in physical space but also in abstract spaces, for example when competition generates patterns in ecological trait spaces [references].”

We decided not to go into the specific topic of clustering, because the questions that are being discussed in this context are not obviously related to the questions addressed in our study.

** Comparison with data and use of phenomenological model.

I appreciate the authors additional text and framing in terms of Pattern-Oriented Modeling. I do also agree that the patterns of the four parameters defined in Eqn 1 change qualitatively with $\log(TP)$ in the same way in both theory and data.

However, I am still finding the motivation for using this parametric form slightly difficult to follow. For example, would it not be cleaner just to evaluate directly total biomass predicted in the theoretical model, and total biomass in the empirical data, and show how these quantities vary with $\log(TP)$? Or to consider maybe some version of a Fourier transform and extracting the dominant frequency, rather than imposing the sinusoidal form?

Total biomass is another conceivable quantitative characterization. However, it is not quite as useful as the characterizations we have chosen, because total biomass can depend sensitively on the upper and/or lower cutoff of the body size range over which biomass is summed. The available data is inconsistent in the size range covered, and the full size range of organisms in these systems is usually not documented. This is why we use the intercepts B_0 of the linear components of the fitted phenomenological models instead.

Our method to capture the periodic modulations is nor much different from doing a Fourier transform and extracting the dominant frequency (just better). The Fourier transform can be derived as the least-square fit of a sum of sin/cos modulations to data.

<https://scicomp.stackexchange.com/questions/11659/least-squares-and-fourier-series> .

The dominant frequency component is the component of the spectrum explaining most of the variation in the data, which is why the least-square fit of only one Fourier component (which adjustable frequency) gives the dominating component. However, the Fourier transform assumes periodicity of the data, and much technical machinery is being used to deal with the fact that real data is rarely periodic. Our direct fit does not suffer from this problem, and we used a median regression rather than least square as a consistent way to deal with gaps in size spectra. The Fourier transform has its uses when data contain a large number of oscillations, in which case the power spectrum can be estimated by smoothing the (absolute value) of the Fourier transform and is then the more reproducible description. In our case there are not sufficiently many oscillation periods in the data for these advantages to materialize.

There may be good reasons for using the parametrized phenomenological model, but readers may need more help to understand these.

We now provide a simple motivation for our choice of the phenomenological model (as explained in our previous response, there is another more technical rationale implicit in the principle of pattern-oriented modelling): “Visual inspection of this data (Fig. 1a, Supplementary Information S4) confirms previous reports [Boudreau, P. R. & Dickie, L. M. Biomass Spectra of Aquatic Ecosystems in Relation to Fisheries Yield. Can. J. Fish. Aquat. Sci. 49, 1528–1538 (1992)] that, on double-logarithm axes, pelagic size spectra exhibit a linear relation that tends to be overlaid with a secondary structure of uniformly spaced domes. To quantify these features, we fitted both empirical and simulated size spectra to a combination of linear relation and sinusoidal modulation of the form [...]”

** One additional question which does not build on my previous review. In Fig 2a a simple demonstration is given of top down control in (some nutrient regimes of) the model. The top dome is depressed, and as a result there is an oscillating (“staircase”) effect on other biomass levels in the community. In fact, for the H value chosen it seems as though the domes are lost altogether, but presumably for smaller H the domes would be present, and diminished in amplitude.

This all makes sense to me. However, one subtlety that may be helpful to clarify. Suppose that I thought (as per literature cited, L49) that “domes represent subsequent members of the aquatic food chain”. Then I would probably look for the effects of top down control in alternating effects on the “heights of the domes themselves”. I.e. the classic staircase pattern that the authors show has upwards steps in the troughs, and downward at the domes. But if I think of the domes as members of a food chain, I am probably expecting the staircase effect to alternate across domes. In simple terms, I am wondering whether readers will expect the cascade arising from top down control to have frequency equal to the frequency of the existing biomass modulation (the authors’ definition) or frequency equal to twice the frequency of the biomass modulation (as per my naive argument above).

This maybe is just another example of where the authors think (and are showing) that the interpretation above of domes as representing members of a food chain is incorrect, or incomplete. Some brief discussion might be informative.

The reviewer offers an interesting and persuasive idea. We now describe this idea briefly in our manuscript (second paragraph of *Basic model behaviour*): “If the domes would represent subsequent members of a food chain, the naive expectation would be that predation release lets the intermediate dome rise when the right-most dome is suppressed, but this is not what we found in simulations.”

We do not say anything stronger, because, as the reviewer remarks, the analogy between food chains and size spectra does indeed break down when looking at the details. In particular, the wavelength of top-down cascades is not twice the wavelength of bottom-up cascades as one would expect if the modes of the latter represented elements of a food chain.

Reviewer #2 (Remarks to the Author):

I believe the authors addressed the comments in a manner that benefited the manuscript and clarified previously confusing points.

At a few places in the manuscript, we edited the text to make it even clearer.

Reviewer #3 (Remarks to the Author):

The revised paper by Rossberg and co-authors presents interesting results and stimulates curiosity. The key result, i.e. the fact that dome patterns in pelagic size spectra would emerge from top-down trophic cascades, is appealing (see fig. 1), even if a better discussion of alternative hypothesis such as outlined in my previous comments (discussion partly provided by the authors as a response to my comments but not included in the paper) would still be needed.

The manuscript discusses in detail the predominant alternative hypothesis offered in the literature, namely that domes are the results of a bottom-up cascade in which subsequent domes represent subsequent members of the aquatic food chain.

The reviewer's comment relates to a third hypothesis, which is (from the reviewer's previous comments):

The domes observed empirically have been shown to match well-identified taxa and functional communities (e.g. viruses, heterotrophic bacteria, unicellular phototrophs, unicellular mixotrophs and heterotrophs, planktonic crustaceans, fish). Each taxa/functional community occupies well-defined and successive size ranges over 3-4 orders of magnitude (e.g. Boudreau and Dickie, 1992; Sprules and Goyke, 1994; Thiebaut and Dickie, 1993; Yurista et al., 2014) due to different limiting processes occurring in these size ranges (Andersen et al., 2016).

Our problems with this hypothesis stem from the fact that, as we noted previously, it is not sufficiently clearly spelled out in the literature cited by the reviewer. Is the claim, for example, that domes form because species corresponding to the size range between domes can not exist at all (this is clearly not the case, because in marine systems domes are weak or absent)? Or is the claim that these taxa just don't exist in the pelagic ecosystems where domes form (in this case, it appears to be rather a description than an explanation of a phenomenon)?

Because we are unsure what exactly the hypothesis is, and because it appears difficult to explain the hypothesis and the supporting evidence in a few words, we are unable to give this hypothesis a fair representation in our manuscript to which we could then respond.

As we previously wrote, the reviewer's idea might well be relevant for explaining the particular positions that the domes attain along the size axis (phase of the dome pattern), **once domes form by some other mechanism**, such as the mechanism that we describe. But this would mean that the reviewer's idea relates to a secondary phenomenon (if it exists): that domes, when they form, tend to attain some specific positions on the size axis.

Demonstration and explanation of this secondary phenomenon (if it exists) would be an interesting topic for a new research project, but this requires that the primary mechanism for dome formation is established first. The present manuscript is concerned with this primary mechanism: why domes form. The manuscript briefly mentions the reviewer's idea in our interpretation as a secondary phenomenon (at the end of "Mechanisms"): "This sensitivity of dome positions to pressures might also lead to a tendency for domes to form in biologically favoured size ranges, for which there is some empirical evidence [23]." We would not consider it prudent to expand on these speculations. The unpublished results that we previously shared with reviewers in this regard are certainly premature, and including them in the manuscript would raise more questions that they would answer.

The prediction that the dome structure increases with the input of nutrients into the system, and therefore mostly appears in eutrophic lakes, also seems to match observations (Fig. 3, S4.1).

However, the model that underpins the entire analysis is not presented in a way that most readers (including the aquatic / marine ecologists and ecosystem modellers that should be the primary target of the paper) could understand. This has not to do with the complexity of the mathematics used; it is a problem of clarity and completeness of the explanations provided as well as a lack of pedagogy. Indeed, almost no information about the model is given in the main paper and the supplementary material is far from being self-standing, despite the addition of fig. S1 (that unfortunately doesn't really help). It is a pity since the results presented would deserve a better justification and the model, which is claimed to be better than previous ones, would also certainly deserve to be presented clearly. Unfortunately this is not the case and at the moment the reader is asked to believe the results without understanding them, and has no choice but accepting the unsupported claims that the SSSM model used in this study is better than previous models in the field and that the secret mechanisms it includes are sufficient to explain the patterns observed.

In response to this feedback we have added a section on *Model structure* in the manuscript. This section explains and motivates the structure of our model by comparing it with related models from the literature, starting with the simplest dynamic models. This explanation includes the discussion of the "correction term" that Reviewer #1 recommended.

But what is different from previous models in the SSSM?

This is now explained in *Model structure*.

What are the mechanisms included that are not included in previous models and how are they included?

As we explain in *Model structure*, the SSSM arises from a simplification of previous size-structured food-web models. No novel mechanisms are added.

Is the model a trait-based model based on species-specific McKendrick–von Foerster equations like trait-based models such as the one in Hartvig et al. (2011) (which is said to be similar to most other recent size-spectrum community models? Why not writing it explicitly? Why the SSSM used here) ...

We took special care to explain in *Model structure* how the species-specific McKendrick–von Foerster Equations enter size-structured food-web models such as that of Hartvig et al. (2011), and why and how they are in our model replaced by an implicit representation of individual-level processes, based on the quasi-neutral approximation.

We now describe our model in the main text as explicitly as is reasonably possible without the use of mathematical notation.

As we explain in *Model structure*, the reason for going over from size-structured food-web models to the SSSM is to recover the formal simplicity and computational efficiency of size-spectrum models.

... don't generate such dome patterns? Is it because they are not integrated over a wide enough size range?

As we explain in the last paragraph of the introduction, many size-spectrum models generate structures that resemble dome patterns. We explain why the predictions by other models differ in decisive details from ours in the penultimate paragraph of *Discussion - A mechanistic explanation*. There we also added a sentence explaining that in case of models of food webs of size-structured populations (Hartvig et al. 2011), the decisive limitation is computational cost. And indeed, as the reviewer suggests, computational constraints are likely to be among the reasons why the latter type of model is not simulated over size range as wide as those considered by us.

Why SSSM don't generate traveling waves while Hartvig's model does?

We could not find any mention of travelling waves by Hartvig et al. (2011). They write about "oscillations", which are not necessarily travelling waves. What Hartvig et al. (2011) demonstrate using their time-averaged data are certainly static waves, similar to those seen in our model. The reason for the "oscillations" seen by Hartvig et al. (2011) are most likely recruitment pulses at species level. These would most probably average out if sufficiently many species were included in their model.

Why showing snapshots and not stationary solutions?

With the correction of our simulation code (previous revision) the model now enters steady states over most of the parameter space. In particular, all solutions shown in Fig. 1 are steady states, and we edited the figure caption to explicitly say so.

Even if the SSSM model is complicated, it has to be presented clearly and pedagogically, in simple terms, introducing the key equations and explaining their relationships without entering technical calculations. This is unfortunately not the case at the moment and the reader is only left with a very vague idea about what the SSSM actually is. Not only is this frustrating, it is also not acceptable in a paper whose central arguments rely on SSSM simulations.

As detailed above, the main text now includes a didactic explanation and motivation of the model structure, including the structure of the underlying equations and how they relate to previous models.

Reviewers' Comments:

Reviewer #1:

Remarks to the Author:

I am satisfied that the authors have addressed my previous rounds of comments on the manuscript.

Reviewer #3:

None